# The Challenges Facing the Current Paradigm Describing Viscoelastic Interactions in Polymer Melts

**DOI:** 10.3390/polym15214309

**Published:** 2023-11-02

**Authors:** Jean Pierre Ibar

**Affiliations:** Polymat Institute, University of the Basque Country (UPV/EHU), 48013 Donostia-San Sebastian, Euskadi, Spain; jpibar@alum.mit.edu

**Keywords:** viscoelasticity, polymer physics, paradigm of polymer rheology, entanglements, Rouse model, reptation model, dual-phase model, grain-field statistics, sustained-orientation, shear-refinement

## Abstract

Staudinger taught us that macromolecules were made up of covalently bonded monomer repeat units chaining up as polymer chains. This paradigm is not challenged in this paper. The main question raised in polymer physics remains: how do these long chains interact and move as a group when submitted to shear deformation at high temperature when they are viscous liquids? The current consensus is that we need to distinguish two cases: the deformation of “un-entangled chains” for macromolecules with molecular weight, M, smaller than M_e_, “the entanglement molecular weight”, and the deformation of “entangled” chains for M > M_e_. The current paradigm stipulates that the properties of polymers derive from the statistical characteristics of the macromolecule itself, the designated statistical system that defines the thermodynamic state of the polymer. The current paradigm claims that the viscoelasticity of un-entangled melts is well described by the Rouse model and that the entanglement issues raised when M > Me, are well understood by the reptation model introduced by de Gennes and colleagues. Both models can be classified in the category of “chain dynamics statistics”. In this paper, we examine in detail the failures and the current challenges facing the current paradigm of polymer rheology: the Rouse model for un-entangled melts, the reptation model for entangled melts, the time–temperature superposition principle, the strain-induced time dependence of viscosity, shear-refinement and sustained-orientation. The basic failure of the current paradigm and its inherent inability to fully describe the experimental reality is documented in this paper. In the discussion and conclusion sections of the paper, we suggest that a different solution to explain the viscoelasticity of polymer chains and of their “entanglement” is needed. This requires a change in paradigm to describe the dynamics of the interactions within the chains and across the chains. A brief description of our currently proposed open dissipative statistical approach, “the Grain-Field Statistics”, is presented.

## 1. Introduction

Staudinger [1] taught us that macromolecules were made up of covalently bonded monomer repeat units chaining up as polymer chains. The chemical nature of the monomer directed the type of covalent bonds conferring most of the specific properties of the polymer. The greater the number of repeat units the longer the chains and the greater the possibility for the chains to assume a variety of shapes, from an extended elongated one to a more compact coiled one. Also, the chemical process that resulted in the synthesis of macromolecules produced many chains, often not with the same shape or size. The properties of the polymers improved when the chains became longer but it was more difficult to process them: their viscosity increased with molecular weight; viscosity was no longer an intensive property like it was for small liquids.

The main question raised in polymer physics was: how do these long chains interact and move as a group when submitted to shear deformation at high temperature when they are viscous liquids? This question is debated in a field of polymer physics called rheology, whose purpose is to understand the viscoelastic aspects of polymer melts deformation [2].

The current consensus is that we need to distinguish two cases: the deformation of “un-entangled chains” for macromolecules with molecular weight, M, smaller than M_e_, “the entanglement molecular weight”, and the deformation of “entangled” chains for M > M_e_. 

Several eminent scientists have extensively studied these two cases over the last 70 years. Paul J. Flory, in 1974, and Pierre-Gilles de Gennes, in 1991, were awarded the Nobel prize in Chemistry and Physics, respectively, for their significant theoretical contribution to the understanding of these challenging problems [3,4]. For both these authors, the properties of polymers derive from the statistical characteristics of the macromolecule itself, the designated statistical system that defines the thermodynamic state of the polymer [5,6]. The molecular weight between entanglements, M_e_, is defined from the rubber elasticity theory and known to be equal to M_c_/2 where M_c_ is the molecular weight for the entanglements when viscosity measurements are carried out. The current paradigm is that the viscoelasticity of un-entangled melts (M < M_c_) is well described by the Rouse model [7] and the entanglement issues raised by the impact of the increase in length of the macromolecules on the melt viscoelasticity, when M > M_c_, are well understood by the reptation model introduced by de Gennes in 1971 [8]. Both models can be classified in the category of “chain dynamics statistics” [9,10,11,12]. 

In this paper, we examine in detail the failures and the current challenges facing the current paradigm of polymer rheology: the Rouse model [7] for M < M_c_, the reptation model [4,6,8] for M > M_c_, the time–temperature superposition principle [13], the strain-induced time dependence of viscosity [14], shear-refinement [15] and sustained-orientation [16]. The basic failure of the current paradigm and its inherent inability to fully describe the experimental reality [17] is reviewed in this paper. 

We focus on re-examining some experimental facts, the most damaging, for these two models based on chain dynamics statistics, being their inability to explain the time dependence of viscosity under small shear strain conditions [14] and the observation of “Sustained-Orientation”, i.e., the reversible triggering of the instability of the network of entanglement [2,16].

In the discussion and conclusion of the paper, we suggest that new concepts are needed to explain the viscoelasticity of polymer chains and of their “entanglement”, also answering a question raised a long time ago [18] regarding their relaxation and thermal analysis behavior. These concepts represent a change of paradigm to describe the dynamics of the interactions within the chains and across the chains. A brief description of our currently proposed open dissipative statistical approach, “the Grain-Field Statistics of Open Dissipative Systems” [19,20,21], is introduced in the conclusion.

## 2. Development

### 2.1. The Great Myth of the Rouse Model: Its Failure to Describe the Rheology of Unentangled Melts (M < M_c_)

#### 2.1.1. (In)validation of the Rouse Model Using Dynamic Data G′(ω), G″(ω)

A classical misconception, already emphasized in other instances ([13], ch. 3 of Ref. [2]), is the statement that polymer melts below Mc follow the predictions of the Rouse model [7]. The myth is so well established that the majority of the authors make this statement without fully verifying the accuracy of the allegation using their own data to validate it.

We give two examples of authors claiming that their data can be fitted by the Rouse model, and present good reasons to dispute such validation. The data both concern dynamic rheological results obtained on a series of monodispersed polystyrene (PS) samples [22,23]. The first set of dynamic data is from Matsumiya and Watanabe (The data were kindly provided by Prof. Watanabe, who also clarified some of the experimental issues by email) [22]. It applies to a monodispersed PS with M = 27,000 obtained at four temperatures T = 115 °C, 120 °C, 130 °C and 140 °C. The second set of dynamic data is from Majeste who studied in his thesis a series of monodispersed PS samples both unentangled and entangled [23]. Note that for Matsumiya and Watanabe’s results the temperatures are all located below the T_LL_ temperature for this molecular weight (164.1 °C, see Equation (18) in [24]), whereas the temperature of T = 160 °C is the reference temperature chosen by Majeste to shift the other frequency sweep isotherms and obtain the mastercurves for eight unentangled PS samples. As we learn in [24], T_LL_ varies with M for PS like T_g_(M) + 70.44 °C, so the choice of T= 160 °C for the mastercurves in Majeste’s data at various M positions the analysis of the data very near below or above T_LL_ for all the molecular weights below Mc. This contrasts with Matsumiya and Watanabe’s data analysis.

The Rouse model is very simple to apply to a set of data: one needs the longest relaxation time, τ_R_, at a given temperature, and the melt modulus G_N_. The melt modulus, G_N_ = ρRT/M, is calculated using the well-known modulus formula taken from rubber elastic theory, where ρ is the melt density, M the molecular weight, and T the value of the temperature (R is the gas constant). In other words, when the molecular weight and the temperature are given, the Rouse model only depends on one parameter, τ_R_. The value of τ_R_ is linearly correlated to the Newtonian viscosity at that temperature, η*_o_; it is also the inverse of the cross-over frequency of G′(ω) and G″(ω), ω_x_, also at the same temperature. The secondary relaxation times, τ_p_ are found from τ_R_: τ_p_= τ_R_/p^2^ with p= 1 to N= M/M_o_ where M_o_ is the mer molecular weight (For PS and M = 27,000 N = 257). A simple spreadsheet permits the calculation of G′(ω) and G″(ω) according to the following Equation.
(1)$$\begin{array}{l} G_{ROUSE\;}^{*}(\omega )=\frac{\rho \mathrm{RT}}{M}\overset{N}{\underset{p=1}{\sum }}\frac{\omega^{2}\tau_{p}^{2}+j\omega \tau_{p}}{1+\omega^{2}\tau_{p}^{2}}with\tau_{p}=\frac{\tau_{R}}{p^{2}}gives\text{:}\;\\ G^{\prime }(\omega )=\frac{\rho \mathrm{RT}}{M}\overset{N}{\underset{p=1}{\sum }}\frac{\omega^{2}\tau_{p}^{2}}{1+\omega^{2}\tau_{p}^{2}}\\ G^{\prime \prime }(\omega )=\frac{\rho \mathrm{RT}}{M}\overset{N}{\underset{p=1}{\sum }}\frac{\omega \tau_{p}}{1+\omega^{2}\tau_{p}^{2}} \end{array}.$$

The density ρ of the PS melts is given by Fox–Flory (Ref. 36 of [23]):(2)$$\frac{1}{\rho }=0.767+5.510^{-4}T+\frac{6.4310^{-2}T}{M}$$

The Rouse time τ_R_ is given by:(3)$$\tau_{R}=\frac{6\eta_{o}}{\pi^{2}}\frac{M}{\rho \mathrm{RT}}$$

The Newtonian viscosity η_o_ is determined at each temperature using the empirical Cole–Cole equation [25] to fit the data log(η*(ω)) vs. logω and extrapolate to ω → 0. For PS M = 27,000, the temperature dependence of the Newtonian viscosity varies with temperature following the Vogel–Fulcher equation [26]:(4)$$\begin{array}{c} \log\left(\eta_{0}(T)\right)=A+\frac{B}{T-T_{\infty }}\\ \;with\;A=-3.20583,B=703.5571,T_{\infty }=44.78^{\circ }C=317.93^{\circ }K \end{array}$$

As already mentioned, the Rouse time can also be determined, τ_R_ = 1/ω_x_, from the cross-over of the Maxwell straight lines passing through the low ω data points of logG′(ω) and logG″(ω) vs. log(ω), by forcing their respective slopes to be two and one in the low ω line regressions, respectively, and calculating the coordinates of the intercepting straight lines.

Let us look at the match between the experimental results of Matsumiya and Watanabe, and the Rouse Equation (1). Figure 1, Figure 2, Figure 3 and Figure 4 compare the results for T = 130 °C.

Figure 1 displays the dynamic moduli G′(ω) and G″(ω) for the data (symbols) and the Rouse Equation (1) (red and blue lines). At first glance, one may say that the fit is remarkably good if one realizes that there is just one fitting constant involved, τ_R_, the Rouse model providing a theoretical basis to determine the other constants G_N_ and τ_p_. The fit is especially good for G″(ω) in the lower frequency region, explaining why the Rouse equation is often validated in the Newtonian range using the viscosity as the variable (G″/ω →η*_o_ at low ω). But, as we have expressed many times ([13], Ch. 3 of [2]), a close examination of the plot makes visible all the objective reasons to reject such a model, which turns out to provide an unacceptable fit of the data. Figure 2 provides the proof.

One of the reasons the apparent fitness of the Rouse model to the data in Figure 1 looked “good”, is that we used log scales on both axes, which clearly compresses the resolution in order to display the overview picture. The log compression of the ω axis covers only three decades of variation of ω, from 0.1 to 100 rad/s. When the curves are mastercurves obtained by horizontal shifting, the log compression extends one-to-three more decades, which makes the appearance of a good fit even better because of the further data compression. Such is the case in the figures presented in Majeste’s thesis, for instance, when they compare the data to the Rouse equation projections [23]. Even in Figure 1, which is not a mastercurve, one can see unacceptable discrepancies when comparing the results: the G′(ω) curves never seem to overlap, a fact proven in the next figure that shows that the residuals for the errors are totally curved when they should be random (i.e., with the points of the residual plot randomly disposed on both sides of the zero horizontal line). Figure 2 provides the % error between the data and its corresponding Rouse prediction. The verdict is crystal clear: there is no range where the fit can be considered acceptable, not even in the low frequency zone, in the terminal region, where Figure 1 gives the illusion of some relative success, especially for G″(ω) as we mentioned earlier. For all the values of ω the residuals are badly curved, the error is two-to-five times the accuracy for measuring the modulus: the Rouse model fails to fit the dynamic behavior for this M < M_c_ melt. This is true for T = 130 °C in Figure 2, as well as for the three other temperatures chosen by Matsumiya and Watanabe (Data not shown). In fact, the errors become much worse for T = 120 and 115 °C. Only T = 140 °C shows a decrease in the magnitude of the errors, yet the residuals are still badly curved.

Figure 3 compares the data and the Rouse dynamic viscosity η*(ω). As in Figure 1, the illusion of a good fit is what is apparent at first, perhaps even more so for the viscosity than for the moduli. All the features of shear-thinning are displayed by the Rouse model: the Newtonian plateau and the decrease in viscosity with strain rate at higher frequency. Yet, these are the same data that produced the unacceptable errors in the determination of G′(ω) and G″(ω) in Figure 2. One sees how the choice of the variables and the use of the log scale can easily mislead the conclusion.

As we said, the elegance of the Rouse model is its lack of fitting constants, being based on a molecular understanding of the motion of a macromolecule to produce flow. The Rouse equation that we have written above can even be further tuned down to include the expression of the radius of gyration of a single macromolecule, R_G_, which can be measured by light or neutron scattering. However, if we desire to optimize the fit between the Rouse’s predictions and the experimental data, we need to make “loose” the value of τ_R_ or G_N_ in Equation (1) and introduce them as regression parameters. The regression fits at low ω become much improved as we do that, yet it is at the expense of the physical Rouse reality: the value of τ_R_ and G_N_ values found by regression become 2000 to 5000% different from their respective values pursuant to the Rouse model (G_N_ = ρRT/M). For instance, if the value of G_N_ is made different for the G′(ω) than for the G″(ω) equation in the Rouse formula (Equation (1)), the fits are considerably improved but the molecular explanation of the model goes down the drain. See below.

Figure 4 is a graph that displays an important apparent discrepancy between the prediction of the Rouse model and the data in the non-Newtonian range of ω. The graph compares the value of χ = (G′/G*)^2^ at various ω either measured experimentally by Matsumiya and Watanabe [22], the black squares, or calculated from the Rouse model (the red dots). What we mean by “discrepancy” is that the large departure between the Rouse model and the data seen in Figure 4 can be demonstrated (as shown below) to not be caused by a transitional high-frequency relaxation process that needs to be introduced to correct the data, it is the demonstration of the failure of the Rouse model to describe shear-thinning correctly. The range of the data investigated in Figure 4 is the lower and middle ω range for shear-thinning, a phenomenon expressing the shear dependence of viscosity, classically exhibited as a departure from the Newtonian range, itself only observed at very low frequency (ω < 10^0^ = 1 rad/s). The reason we bring this up is to differentiate our conclusions about the origin of the differences (observed at higher frequency in Figure 4) as a failure of the Rouse model, from the explanations by many other authors, such as Majeste in his thesis, who have claimed, that the Rouse model basic equations can be corrected to include the influence of the transitional high frequency relaxation terms on the dynamics of flow, thus would have attributed the differences in Figure 4 to the lack of corrections pursuant to the transitional high-frequency relaxation terms. We dedicate the following section to disprove the applicability of these authors’ argument.

We have expressed in several previous publications [2,14,27] our interest in the variable χ = (G′/G*)^2^, equal to cos^2^δ, also equal to 1/(1 + tan^2^δ), where δ is the phase angle between stress and strain. This parameter χ, we have suggested, is more appropriate than other traditional rheological variables (such as G″ or tan δ) to describe the viscoelastic character of the melt, especially when it is formulated in terms of the Dual-Phase and Cross-Dual-Phase parameters [20]. The maximum of χ(ω), visible in Figure 4, corresponding to a minimum of tan δ, is known to occur for entangled (M > M_c_) melts, and its frequency occurrence is attributed to the beginning of the rubbery plateau. In the case of unentangled melts, however, such as is the case for the sample of Watanabe and colleagues in Figure 4, the current paradigm understanding is that there is no rubbery plateau and therefore the phenomenon giving rise to the maximum in Figure 4 must have a different origin than the onset of entanglements. Since the absence of the rubbery plateau implies the juxtaposition of the terminal region and the T_α_ transitional region, many authors were led to attribute the departure they saw in their higher frequency data to the presence of the transitional high-frequency relaxation terms, the so-called T_α_ terms. Note that the Rouse model is not capable, on its own and without correction, of making χ exhibit a maximum (or a minimum of tan δ). The simple reason is that, in the Rouse mathematical formulation limited to τ_R_, χ is equal to G′/G_N_, (see Equation (16) of Ref. [13]). Since its G′(ω) never exhibits a maximum for all molecular weights and all values of the frequency ω, the Rouse model is doomed to fail to explain the maximum in Figure 4 without adding at least an extra term.

This failure is, indeed, recognized by the molecular models of the unentangled state which have considered correcting the Rouse modulus to include an extra term due to these high-frequency relaxations. Likewise, earlier models than the dynamic molecular Rouse’s model that tackle viscoelastic network deformation by expressing the moduli in terms of a spectrum of relaxations have shown the need to correct the high-frequency terms. For instance, Gray, Harrison and Lamb [28] considered a continuous and dissymmetric distribution of the relaxation times of the type Davidson and Cole [29] resulting in the modification of the complex compliance to include three terms. This manipulation of the spectrum of relaxation did result in a very good fit between the data and the corrected deformation model, such as applied to Rouse [25], but amounts to modify-to-fit the spectrum of relaxation without a sound physical foundation to justify it. The use of mathematical patches to make failing models fit the results may be useful if they point to the right direction to what needs to be done theoretically to modify the initial assumptions of the model. In the case of models based on the spectrum of relaxation profile, the Gray et al.’s corrections of the spectrum of relaxation represents a real success. In the case of the Rouse model, we have quoted in Ref. [13] (Equation (18)) an expression due to Allal [30] that has been claimed to extend the range of fitness of the Rouse expression of G′(ω) and G″(ω) to higher frequencies. Majeste used Allal’s method to correct his data and claimed that it improved the fits to the Rouse model [23]. We evaluate in detail below the merit of such improvements and its relevance to explain the discrepancy in Figure 4.

Equation (5) explains the Allal’s high-frequency correction which adds a new term, G*_HF_, to the complex modulus.
(5)$$\begin{array}{l} Modulus\;according\;to\;Rouse:\\ \begin{array}{cc} & G_{ROUSE}^{*}(\omega )=\frac{\rho RT}{M}\sum_{p=1}^{N}\frac{\omega^{2}\tau_{p}^{2}+j\omega \tau_{p}}{1+\omega^{2}\tau_{p}^{2}}\\ & \tau_{p}=\frac{\tau_{R}}{p^{2}}\\ & G^{*}(\omega )=G^{\prime }(\omega )+jG^{\prime \prime }(\omega ) \end{array}\\ For\;the\;High\;Frequency\;terms:\\ \begin{array}{cc} & G_{HF}^{*}(\omega )=G_{\infty }\left[1-\frac{1}{\sqrt{1+j\omega {\tau^{\prime }}_{o}}}\right]\\ & {\tau ’}_{o}=\frac{1}{\pi^{2}}\frac{\xi_{o}b^{\prime 2}}{kT} \end{array}\\ Total\;Modulus:\\ G^{*}(\omega )=G_{ROUSE}^{*}(\omega )+G_{HF}^{*}(\omega ) \end{array}$$

In this equation, G_∞_ is the glass modulus, ξ_o_ the monomeric friction coefficient, b’ the monomeric length, j the imaginary unity number (j^2^ = −1) and k is the Boltzmann’s constant (1.38065 × 10^−23^). We have found two sets of values for the molecular parameters introduced in Equation (5): ξo, b′ and G_∞_. Leonardi (Table II-1 of Ref. [31]) studied a PS with M_w_ = 326,000 and polydispersity I = 3.4 and gives the following values: ξ_o_ = 6.3 × 10^−8^ Kg/s, b′ = 7.4 × 10^−10^ m and G∞ = 6 × 109 Pa. This PS sample is entangled and polydispersed. For T = 130 °C (i.e., 403 °K in Equation (5)), the value of τ′_o_ is: 6.2823 × 10^−7^ s. Majeste [23] studied 8 monodispersed unentangled PS samples and provides for those grades the following values: ξ_o_ = 2.7 × 10^−14^ Kg/s, b′ = 3.7 × 10^−10^ m and G_∞_ = 1 × 10^9^ Pa. For T = 130 °C we now find τ′_o_ = 6.7310 × 10^−14^ s. This high-frequency relaxation time is one million times smaller than that found for the Leonardi’s entangled PS. It is unclear why the fundamental molecular parameters entering the Allal’s high-frequency relaxation correcting term would make τ′_o_ vary significantly with the length and the polydispersity of the chain. Could it be a new characteristic of entanglements? The physical reason for such a huge variation of τ′_o_ appears doubtful because the high-frequency component is supposed to represent the local relaxation at the monomeric level and should not depend, at least approximately, on the length of the chain, whether it is entangled or not. Such large differences in the values of ξ_o_ and b′ between the two PS samples of Majeste and Leonardi do not make sense. Additionally, assuming that the value of ξ_o_ and b′ tabulated by either Majeste or Leonardi are acceptable, we have found another reason to be concerned with the Allal’s formulation of the Rouse’s correction and it is exposed below.

G*_HF_ is a complex number in Equation (5) that can be decomposed into an elastic and viscous component by way of the de Moivre’s formula to remove the square root:(6)$$\begin{array}{r} G_{HF}^{\prime }=G_{\infty }\left(1-\sqrt{\rho_{HF}}cos\left(\theta_{HF}/2\right)\right)\\ G_{HF}^{\prime \prime }=-G_{\infty }\sqrt{\rho_{HF}}sin\left(\theta_{HF}/2\right) \end{array}\;\mathrm{with}:\;\begin{array}{r} \rho_{HF}=\frac{1}{\sqrt{1+\omega^{2}{\tau^{\prime }}_{o}^{2}}}\\ tan\left(\theta_{HF}\right)={-\omega \tau^{\prime }}_{o} \end{array}$$


Figure 5 is a graph of G′_HF_(ω) and G″_HF_(ω) versus logω for T = 130 °C using the ξ_o_, b′ and G_∞_ constants of Leonardi plugged into Equations (5) and (6). The data range region covers the ω range between 0.1 and 100. The higher values of ω are added to show how Allal’s formula works. In this figure, in the data range region, the magnitude of the value found for G″_HF_(ω) is greater than its G′_HF_(ω) counterpart by more than five decades! When we add these high-frequency correction moduli, G′_HF_(ω) and G″_HF_(ω), to the Rouse modulus G′(ω) and G″(ω), respectively, using the PS M = 27,000 sample, we observe that G′(ω) remains unchanged because G′_HF_(ω) is irrelevantly small, and that the corrected G″(ω) is worse than the uncorrected G″(ω) Rouse modulus in the low ω region, the very region where the fit to the real data was not so bad. This is shown in Figure 6, a plot of G″(ω) = G″_ROUSE_(ω) + G″_HF_(ω) , using the Leonardi’s parameters, versus G″(ω) data. The Rouse modulus corresponds to the red dots, the corrected Rouse modulus is represented by the black square, the perfect fit is the straight line y = x. One sees that the black squares are further away from the perfect fit. The conclusion is that the Allal’s high-frequency correction appears to render the fit worse than the pure Rouse equation: the Rouse correction cannot be applied to the PS 27,000 data using Leonardi’s PS molecular parameters. The other possibility is that only the molecular constants of a monodispersed unentangled PS should be used in the Allal’s equations when applied to PS 27,000 which is also unentangled and monodispersed. Figure 7 is the same graph as Figure 5 but uses the molecular parameters assumed by Majeste, everything else being the same. One sees that the correction moduli, G′_HF_(ω) and G″_HF_(ω), are now both too small to add anything relevant to the values of the uncorrected Rouse moduli in the data range used to analyze this polymer. This is confirmed in Figure 8, similar to Figure 6 but now using the Majeste’s constants in the Allal’s equation. In Figure 8, the graphs before and after correction are identical and still very poorly fitting the data.

In conclusion, the Allal’s high-frequency terms added to the Rouse terms does not improve fitting the data. The use of the Rouse formulation should be limited to the Newtonian (terminal) region and is not adequate to describe shear-thinning of unentangled polymer melts.

Note that Majeste applied the Allal’s corrections to the mastercurves obtained after shifting the isotherms to T = 160 °C. This shifting expands by a couple of decades the span of frequency toward the higher frequency region. Yet, looking at Figure 7 and expanding the data range to the right by two or three decades will not increase the values of G′_HF_(ω) and G″_HF_(ω) sufficiently to explain the large residuals observed in Figure 2. In addition, the time–temperature superposition principle needs to be validated over the data range analyzed in order to apply it with confidence, and, as will be shown below, the time–temperature shifting validity is limited for the Matsumiya and Watanabe’s PS to the low ω range, below the maximum of χ vs. logω in Figure 3.

The reason we conclude that the Allal’s high-frequency term has nothing to do with the maximum of χ observed in Figure 3, which occurs around ω = 10 rad/s, is the value of τ′_o_ in the expression of G*_HF_ in Equation (5). With the values provided by Leonardi, τ′_o_ is around 10^−6^ to 10^−7^, which, we agree, is “big enough” to start to have an impact in the ω = 0.1 to 100 rad/s range. However, we saw that the parameters provided for the Leonardi’s PS made the Allal’s corrections worse, not better. For the other option, with τ′_o_~10^−14^ to 10^−13^, the values provided by Majeste, corresponding to the vibrational motions in the glassy state, it is clear that Allal’s correction G*_HF_(ω) will practically remain equal to 0 until the frequency is near resonance, i.e., until ω~1/τ′_o_.

In conclusion, although the basic idea of adding a “glassy component” to the behavior in the terminal region makes sense to try to complement the Rouse’s basic molecular dynamic contribution, it does not appear to be correctly addressed by Allal’s formula. In addition, as we will show using the data of Matsumiya and Watanabe, the time–temperature superposition principle only applies satisfactorily within a limited range of temperature, which raises some questions regarding the accuracy of the shifted data at high frequency in the case of the Majeste’s data.

In conclusion, the “satisfying improvement of the fitting of the data” claimed by Majeste to be the result of adding the high-frequency Allal’s correction is definitely overstated, to say the least. For instance, Figure 1.88 (M = 8500) of Majeste’s thesis (not reproduced here) clearly shows that the improvement is not satisfactory, according to our standards: all the calculated G′(ω) values calculated after corrections are systematically off the data values, even in the terminal region, and using log–log axes. This seems to be the same type of fitting failure observed for G′(ω) in Figure 1 for the M = 27,000 PS.

#### 2.1.2. (In)validity of the Rouse Formula to Predict the Molecular Dependency of τ_R_ below M_c_

We said that the Rouse model was not capable of describing well the dynamics of shear-thinning for unentangled melts, but also added that its use could be limited to the Newtonian region. Does this mean that the Rouse model is correct/useful in the terminal regime of viscoelasticity? This is what we want to examine in this section.

Equation (3) is often used to validate the Rouse model. There are two ways to verify this formula, one at T constant, M variable, i.e., using the Majeste’s data at T = 160 °C, and by working at M constant and varying T, i.e., using Matsumiya and Watanabe’s data at M = 27,000 and T variable between 115 and 140 °C.

As we already mentioned before, the Maxwell lines cross-over, ω_x_, can easily been found from fitting the low ω region where G′(ω) and G″(ω) can be forced to verify the Maxwell’s slopes of 2 and 1 when plotted against ω on log–log axes. This provides an accurate way to determine τ_R_=1/ω_x_.

##### Molecular Weight Dependence of the Rouse Time, τ_R_ from the M < M_c_ Majeste Data

The Rouse theory implies that τ_R_ is proportional to M^2^, which is equivalent to predicting that η_o_ is proportional to M.

Figure 9 is a plot of log ω_x_ vs. logM for the 8 unentangled samples of Majeste at T = 160 °C. We expect to see a slope of −2 if the Rouse formula is validated.

The graph in Figure 9 is better described by two straight lines than by just one. The crossing of these two lines occurs for log M~3.67 (M = 4700) that we have designated M′_c_. The regression line passing through the points M > M′_c_ has a (forced) slope of −3.0 ± 0 and an intercept equal to 15.92 ± 0.033 (r^2^ = 0.987, χ^2^/DoF = 0.0067). This regression straight line is the red line drawn in Figure 9 passing randomly through six data points including M′_c_. The unconstrained slope, −2.92 ± 0.16 (r^2^ = 0.994), also points towards a slope of −3.

The conclusion is that the slope is not equal to −2 as it should be if the Rouse formula had been validated. It is clear that a slope of −3, although unexpected, is closer to what is measured (−2.92). In such a case, η_o_ would vary against M^2^ instead of M. The other observation concerns M′_c_ that we find at approximately M = M_c_/8 (4375). This same M′_c_ “transition” in the log viscosity–logM curve is observed using viscometry data on the same monodispersed PS, but this is not the subject of this presentation. Needless to say, the Rouse model does not predict the presence of M′_c_. We let the reader know, in that regard, that in our Dual-Phase model of polymer interactions to describe viscoelasticity [20], the transition M′_c_ is predicted and represents the molecular weight for the formation of stable macrocoils, the first rheological manifestation of the macromolecular aspect of the interactive systems of mers.

##### Temperature Dependence of τ_R_ at M Constant

Equation (3) involves the Newtonian viscosity, η_o_, the shear elastic modulus of the melt, G_N_ = ρRT/M, and τ_R_ computed from the cross-over frequency: ω_x_ = 1/τ_R_. G_N_ is calculated by Equation (2) and the Newtonian viscosity by Equation (4). All these variables are temperature dependent and known. We can test its validity by plotting the product G_N_ τ_x_(T) versus η*_o_(T). The Rouse equation is validated if the slope is equal to 6/π^2^ = 0.608

A linear regression applied to the 3 upper isotherms, T = 140 to 120 °C, is represented by the red segment in Figure 10. It is a quasi-perfect linear fit with r^2^ = 1.0; the slope, 0.6151, is almost exactly what is projected by the Rouse model (0.60935). The extrapolation is slightly off the T = 115 °C data point at the top of the figure, but this offset is expected and will be explained in the next section.

In other words, the Rouse model correctly describes the relationship between η_o_(T) and τ_R_(T) at M constant and correctly assigns the ratio of the viscosity to the relaxation time (G*/τ_R_) to the melt modulus: (6/π^2^) G_N_. **This result is not a minor achievement of the Rouse model.** This correct prediction of G_N_(T) may explain its popularity at a time when the relaxation processes in polymers were mainly described by networks of spring and dashpots placed in series (Maxwell network) or in parallel (Voigt network). In these networks, the relaxation time was equal to the ratio of the spring modulus to the dashpot viscosity (τ = G/η).

Yet, the validation of Equation (3) provided by Figure 10 corresponds to validating G_N_ from G″(ω) in Equation (1) since η*_o_ = lim (G″/ω) when ω → 0. Equation (1) assumes that G′(ω) and G″(ω) have both the same terminal relaxation time, τ_R_, and the same normalization modulus, G_N_. Equation (3) can easily be derived from the G″(ω)/G_N_ side of the Rouse formula:(7a)$$G^{\prime \prime }{\;}_{\mathrm{ROUSE}}(\omega )=G_{N}\sum_{p=1}^{N}\;\frac{\omega \tau_{p}}{1+\omega^{2}\tau_{p}^{2}}\;\mathrm{when}\;\omega \to 0\;\mathrm{the}\;G_{\mathrm{Rouse}\;}^{\prime \prime }(\omega )\;\mathrm{simplifies}\;\mathrm{to}:\;G_{\mathrm{ROUSE}\;}^{’’}(\omega )=G_{N}\sum_{p=1}^{N}\;\frac{\omega \tau_{p}}{1}=\omega G_{N}\tau_{R}\sum_{p=1}^{N}\;\frac{1}{p^{2}}\;\mathrm{from}\;\mathrm{which}\;\mathrm{we}\;\mathrm{derive}:\;\frac{G^{\prime \prime }}{\omega }=\eta_{o}=G_{N}\tau_{R}\sum_{p=1}^{N}\;\frac{1}{p^{2}}\;\mathrm{For}\;M=27,000\;g/\;\mathrm{mole}\;\mathrm{and}\;M_{o}=105\;g/\;\mathrm{mole}\;N=259\;1+\frac{1}{4}+\frac{1}{9}+\frac{1}{16}+\ldots +\frac{1}{259^{2}}=1.6410805\;\mathrm{For}\;N\to \infty ,\;\mathrm{the}\mathrm{sum}\mathrm{equals}\;\pi^{2}/6=1.644926=0.60793^{-1}\;\mathrm{Hence}:\;\tau_{R}=\frac{\eta_{o}}{G_{N}}\frac{1}{\sum_{p=1}^{N}\;\;\frac{1}{p^{2}}}=0.60935\frac{\eta_{0}}{G_{N}}∼0.61\frac{\eta_{0}}{\frac{\rho RT}{M}}$$

The last line of Equation (7a) is Equation (3) verified by Figure 10. In other words, G″(ω) is correctly normalized by the rubber elasticity theory modulus G_N_ when ω is in the Newtonian range.

We now need to check that the G′(ω)/G_N_ part of the Rouse formula in Equation (1) is also validated in the Newtonian range at ω → 0.
(7b)$$G_{\mathrm{ROUSE}}^{\prime }\left(\omega \right)=G_{N}\sum_{p=1}^{N}\frac{\omega^{2}\tau_{p}^{2}}{1+\omega^{2}\tau_{p}^{2}}=\omega^{2}\tau_{R}^{2}\left[\sum_{p=1}^{N}\frac{1}{p^{2}+\omega^{2}\tau_{R}^{2}}\right]\;with\;G_{N}=\frac{\rho RT}{M}\;when\;\omega \to 0\sum_{p=1}^{N}\frac{1}{p^{2}+\omega^{2}\tau_{R}^{2}}\to 1.64108\;and:\frac{G^{\prime }{}_{\mathrm{ROUSE}\;}\left(\omega ,T\right)}{G_{N}}=1.64108\omega^{2}{\tau_{R}}^{2}\;\mathrm{Hence},\;\mathrm{in}\;\mathrm{the}\;\mathrm{Newtonian}\;\mathrm{regime}\text{:}\;\mathrm{lo}g\left(G^{\prime }\right)=0.21513+2\mathrm{lo}g\left(\tau_{R}\right)+\mathrm{lo}g\left(G_{N}\right)+2\mathrm{lo}g\left(\omega \right)\;\mathrm{Intercept}\;\mathrm{of}\;\mathrm{Maxwell}\;\mathrm{straight}\;\mathrm{line}\;\mathrm{relative}\;\mathrm{to}\;G^{\prime },I_{M}^{\prime }:\;I_{M}^{\prime }=0.21513+2log\left(\tau_{R}\right)+\log\left(G_{N}\right)\;\mathrm{so}\text{:}\;log{\left(G_{N}\right)}_{\exp\;}=I_{M}^{\prime }-0.21513+2log(\omega_{X})$$

The testing of the Rouse equation in the Newtonian regime conducted from a G′(ω) point of view can be done by plotting first log G′(ω) vs. logω and fitting the low ω range with a Maxwell’s straight line of slope 2. This is shown in Figure 11 for the M = 27,000 PS of Matsumiya and Watanabe. at T = 130 °C. The intercept is I’_M_ = 4.6973. The value of ω_x_ to calculate τ_R_, ω_x_ = 1/τ_R_, imposes itself because τ_R_ is the same for G′(ω) and G″(ω) in the Rouse equation and thus ω_x_ must be at the cross-over point where G′(ω_x_) = G″(ω_x_). When ω_x_ and I’_M_ are known, Equation (7b) provides the value of G_Nexp_ calculated from G′(ω). We repeat the same operation for the other temperatures that show a Maxwell’s behavior at low ω, i.e., for T = 120 °C and 140 °C. The T = 115 °C plot, similarly to what we saw in Figure 11, does not present a range of data points that could be fitted by a straight line with slope 2 in the low ω data range). For the Rouse model to be validated we should have G_Nexp_ = G_N_ = ρRT/M: so, if we plot G_Nexp_(T) vs. G_N_(T), we should find all the points on the line Y = X. The values of ω_x_, G_N_, I′_M_ and G_Nexp_ are confined in Table 1.

Figure 12 demonstrates the clear failure of the Rouse model to predict in the Newtonian region the correct G_N_ value that normalizes the G′(ω) moduli data of Matsumiya and Watanabe. We can draw a straight line passing through the three data points in Figure 12 and a linear regression gives: G_Nexp_ = (−1.39 × 10^6^ + 11.9 G_NRouse_) with r^2^ = 0.999. One sees that G_N_ calculated from G′(ω) is not equal to G_N_ calculated from G″(ω) and, therefore, the validation of the Rouse model that emerged from Figure 10 based on the G″(ω) Newtonian branch of the Rouse Equation (1) is contradicted without ambiguity by Figure 12. The only temperature at which the two G_N_ values coincide is at the crossing of the Y = X and the red line in Figure 12, occurring for G_N_ = 127,500, which, according to G_N_(T) occurs at T = 137.5 °C. We cannot predict, without the necessary experimental data to test it, whether the invalidation of the Rouse model persists at higher temperature, for instance above T_LL_~164 °C for this polymer.

We saw in Figure 9 that the Rouse model failed to describe the molecular dependence of ω_x_ when Equation (3) was applied to the Majeste data at T = 160 °C. This meant to say that the Rouse modulus, G_N_, although good to predict G_N_(T) when calculated from G″(ω), was not good to predict G_N_(M). What about the value of G_N_(M) found from G′(ω): does it match the Rouse model molecular modulus, G_N_ = ρRT/M with ρ(M) given by Equation (2)?

##### Comparing the Calculations of G_N_ from the G′(ω) or G″(ω) Sides of the Rouse Equations and Invalidating the Rouse Approach

In order to proceed with this new (in)validation step, we find for each unentangled PS sample of Majeste’s thesis the value of I’_M_ given by Equation (7b) by plotting log G′(ω) vs. logω. This is illustrated in Figure 13 for M = 13,000. We already know the values of ω_x_ for all these samples (Table 2) and thus can calculate the value of log G_Nexp_ (M) for each M and compare it with the corresponding value of G_N_ from the Rouse equation. This comparison is done in Figure 14. Finally, Figure 15 compares the log G_N_ vs. log M at T = 160 °C for the Majeste PS unentangled samples with G_N_ coming from three sources: from the Rouse Equation (black squares), from Equation (7a), i.e., the G″(ω) data based on ω_x_ and η_o_ (blue triangles), and from Equation (7b), i.e., the G′(ω) data based on ω_x_ and I′_M_ (red dots).

#### 2.1.3. Conclusion Regarding the Myth of the Applicability of the Rouse Equation to the Rheology of Unentangled Polymer Melts

The results of our investigation regarding the applicability of the Rouse model are provided in Table 1 and Table 2. These results and the figures drawn from them are devastating for the Rouse model’s validation to describe polymer melts. Its failure to describe experimental data is so flagrant and demonstrated in so many ways that it is almost incomprehensible that both authors of the data which we re-analyzed concluded that the Rouse model satisfactorily described their data. Take Figure 15 for instance, the black squares on the straight line are the points given by the Rouse formula for G_N_, the melt modulus. The red dots and the blue triangles are calculated from the Rouse equation (Equation (1)) taking either the G′(ω) or the G″(ω) expressions in the Newtonian range to find G_N_, respectively. These red dots and blue triangles should all be disposed on the black line if the Rouse theory was applicable to these data. What we observe, instead, is unambiguously different: the red dots are all located above the black line, shifted vertically by almost a decade and a half and the blue triangles are all scattered below the black line, almost forming a straight line pattern. The myth of the Rouse model applicability to unentangled polymer melts is so anchored in the current paradigm that even the most reliable polymer scientists fail to test it fully on their own data.

An important conclusion of the failure of the Rouse model to satisfactorily describe rheological data for unentangled polymer melts is that the normalizing melt modulus, G_N_, is different for the elastic and the viscous components, G′(ω) and G″(ω) of the complex modulus, G* = G′ + jG″. We can call them G′_N_ and G″_N_, respectively. We could also push this exercise one step further and consider that the elastic and viscous components of G* have different terminal times, say τ′_R_ and τ″_R_, respectively. The Rouse equation remains the same, formerly, but we have made the real and the imaginary terms of the complex function G* “independent”. These two terms might still be coupled but in a way different than what is implied by the un-modified Rouse equation. We have already mentioned above that making the G_N_ and τ_R_ constants “loose” in the non-linear regression of the Rouse formula improved the fit of the data a great deal. We also added, however, that this bifurcation from the Rouse basic formula inevitably took us away from the Rouse molecular reality. But what is the real molecular reality? Is the modulus of the melt truly a complex entity with elastic and dissipative coordinates? The Rouse equation establishes very simply the viscoelastic nature of the melt by considering the formula of two functions G′(ω,T) and G″(ω,T) and stating that they are the real and imaginary coordinates of a complex function. This mathematical foundation—based on the coupling between a spring and a dashpot in a mechanical system—correctly led to the observed Maxwell slopes of 2 and 1 for the log G′(ω) or log G″(ω) when plotted against logω in the Newtonian region. The true appeal of the Rouse model is to have found a molecular basis for the “spring and dashpot” mathematical parameters, G_N_ and τ_R_. The problem of the Rouse model is illustrated in Figure 15 that shows that beautiful and simple mathematics might be enough to create a myth but not enough to be validated through its confrontation with the data. We show in Chapter I.5 of Ref. [20] (simulation of the Dual-Phase model) that the attribution of the Maxwell slopes 2 and 1 is not necessarily derived from a Rouse type of mathematical formalism, and that the origin of and the coupling between G′(ω), that force proportional to ω^2^, and G″(ω), that force proportional to ω, may be understood in a way fundamentally different than a complex dependence of the type: G* = G′ + jG″.

In conclusion, there is no merit to the Rouse model, in our opinion, in its present formulation, Equation (1). The Rouse model fails to describe the viscoelastic behavior of unentangled polymeric melts. The introduction of G_N_ = ρRT/M, borrowed from the rubber elasticity theory, permits to normalize the dynamic moduli, G′(T,ω) and G″(T,ω), but it has no molecular meaning in the Rouse’s physical reality (or if it does, the theory of rubber elasticity must be reconsidered). The introduction of τ_R_, the Rouse time, since it is the inverse of the cross-over frequency, τ_R_ = 1/ω_x_, is useful, practically: it permits the introduction of a “marker” of the state of the melt, more or less correlated to the end of the Newtonian range for ω. Of course we do not need a theoretical meaning to use τ_R_, and there is none. The physical molecular modelization of flow proposed by the Rouse model is wrong: like any molecular model that considers the single chain as the system to explain the flow behavior it cannot predict the existence of any transition in the melt, neither the T_g_ + 23 °C transition [18,19] nor T_LL_ (see next section). The temperature and molecular weight dependence of τ_R_ = 1/ω_x_ also provide useful information. However, τ_R_ is totally useless (theoretically) to quantify the “dynamics” of the viscoelastic behavior, i.e., shear-thinning, the effect of rate and temperature on the kinetics of molecular motion. The spectrum of relaxation generated by τ_p_= τ_R_/p^2^ is simple but useless to correctly describe shear-thinning or to understand why χ(ω) = (G′/G*)^2^ presents a maximum. Likewise, we find the various attempts to modify the Rouse equation by either “truncating-to-fit” the spectrum of relaxation or by adding a high frequency term to the Rouse modulus (the Allal’s approach) to be either empirical or not working according to the claims (despite of our best efforts to make these attempts work).

### 2.2. The Myth of the Extended Applicability of the Time–Temperature Superposition Principle

The “time–temperature superposition” principle is an extrapolation method that permits to extend the range of measurement of an experiment, in time or in frequency, by operating at other temperatures and shifting the multi-T data to obtain a mastercurve, at a given chosen temperature, with the extended time/frequency range. This extrapolation method has been extensively used, for instance, to present the full view of the elastic modulus from the molten state temperature region to the glassy state region, a complete picture that no single instrument can provide. It is, therefore, of the utmost importance to trust the method of extrapolation in question, namely the validity of the Time–Temperature superposition claims.

The current paradigm of polymer physics teaches that the validity of the “time-temperature superposition principle” (tts) covers the range T_g_ to T_g_ + 100 °C, i.e., works approximately over a 100 degrees range above T_g_ [32]. The tts is also applicable to dynamic data obtained by frequency sweeps at constant temperature, i.e., under oscillation at various frequencies ω under given temperatures. The tts expresses the following: the rheological variables found at temperature T_1_, using frequency ω_1_, are the same as those found at T using frequency ω provided the time scale (here the frequency) is changed by a shift factor, log a_T_ =log(ω/ω_1_), which varies with temperature only; the Vogel-Fulcher equation (see Equation (4) can be re-arranged to describe log a_T_ as a function of T and T_1_ (WLF equation, Ch. 11 of Ref. [32]). The moduli to superpose must be normalized by the Rouse modulus, G_N_ = ρRT/M, before superposition. This amounts to say that there is a vertical shift factor b_T_ = ρ_1_T_1_/ρT to be applied to the modulus variables to superpose in order to optimize the superposition. The data set at T_1_, ω_1_ is called the reference data set; the other data sets at T, ω are shifted by b_T_ on the vertical axis and a_T_ on the horizontal axis to produce a mastercurve at T_1_. We have discussed the limitations of the tts and its status as a myth in Ch. 3, pp. 59–73 of Ref. [2] and we refer to that writing for more details. To summarize our findings:The superposition of curves by horizontal shifting on the log time or log frequency is a good approximation over a rather short temperature interval. There are 3 ranges of temperature within which the tts works well for polymer melts: the (T_g_ to T_g_ + 23 ± 2) region, the (T_g_ + 23) to T_LL_ region and the T > T_LL_ region. For each temperature region a new set of WLF constants (or Vogel Fulcher constants) must be established. Superposition across regions is physically improper according to the Dual-Phase model [20].The use of b_T_ pursuant to the normalization of the moduli by the Rouse modulus G_N_ is incorrect. The reason has been implicitly given in the previous section which showed the inadequacy of using G_N_ except for G″(ω) and thus viscosity (Figure 10). To find the correct value of b_T_, a double-shifting regression is always required [33]. It has been shown, for instance, that the vertical shift factor, log b_T_, when it is obtained by regression-double-shifting, is not as predicted by the Rouse modulus G_N_/G_N1_, yet that its variation with T permits to detect the presence of transitions, such as the transition at T_g_ + 23 °C also visible from thermal stimulated depolarization data [19] or the T_LL_ transition [24,34].The temperature range of applicability of the tts varies with the strain imposed during the frequency sweeps ([2] “Effect of Strain” (Section 5.8, p. 322) and with the thermal-mechanical history of the melt prior to the frequency sweep ([2] “Thermal-Mechanical History to create out-of-equilibrium melt properties”, Section 4.3.5.2 p. 206).The tts might be valid for a limited frequency range only or it might be valid on two or successive frequency ranges with different constants to express the 2 shift factors, log a_T_ and log b_T_. It is the case for the 3 temperature ranges delimited by T_g_ + 23 and T_LL._

In this section we want to illustrate the difficulty encountered applying the frequency-temperature superposition to the data of Matsumiya and Watanabe. already introduced in the previous section. These data on a well characterized monodispersed PS are within the range of temperature above T_g_ (T_g_ = 93.78 °C for M = 27,000) where the time–temperature superposition is claimed to apply, and, the range of temperature analyzed is only 25 °C (from T = 115 °C to 140 °C). The melt is located below its T_LL_ evaluated at 161.4 °C for M = 27,000. Also note that T_g_ + 23 °C = 116.78 °C, which positions T = 115 °C inside the T_g_ + 23 range (barely though) and T = 120, 130 and 140 in a different range, the (T_g_ + 23) to T_LL_ range. Our intention is to show that the principle of superposition does not work well for these data because it needs to be perfected based on a better understanding of its origin and its limitations. The possible reasons for the need to modify and limit the time–temperature superposition naturally shift the light on the necessity to reconsider our understanding of the physics of the interactions in polymers. A quantitative explanation of the rheological results of Matsumiya and Watanabe. based on the concepts of the new paradigm is described in another publication (Ch. II.7 of Ref. [20]) and not in this paper.

The time–temperature principle is illustrated in Figure 16, Figure 17, Figure 18, Figure 19, Figure 20, Figure 21, Figure 22, Figure 23, Figure 24 and Figure 25 using the data of Matsumiya and Watanabe which are obtained by dynamic rheometry. These authors have described their experimental procedure as follows: the frequency sweeps were “DOWN sweeps”, from high to low frequency (100 to 0.1 rad/s). The temperature for the 1st sweep was 140 °C, followed by the other frequency sweeps done at the lower temperatures (130, 120, 115 °C in this order) using the same sample. The strain was chosen to keep the results in the linear viscoelastic region (2%). This procedure is not unusual but is different from the one used most often that consists of UP sweeps and changing the sample after each frequency sweep to avoid the slightest possibility of inducing a thermal-mechanical history in the sample when operating sequentially on the same sample even in the linear range.

Figure 16 and Figure 17 are plotted from the original data of Matsumiya and Watanabe which were kindly provided to this author. The black squares represent the “reduced” modulus values, i.e., G′ and G″ corrected by T_1_/T where T_1_ = 115 is the reference temperature and T is the temperature of the frequency sweep to shift, both converted to °K. This correction is induced by the adherence to the Rouse model for which the dynamic moduli are proportional to G_N_ = ρRT/M (Equation (1). The round red dots (reduced in size to avoid overlapping the black squares) are the data without any temperature correction. The difference between the red dots and the black squares is hardly visible. The small temperature interval (25°) renders the Rouse correction of the moduli negligible.

The complex viscosity, η*(ω) = G*/ω, is calculated from the values of G′(ω) and G″(ω) in Figure 16 and Figure 17, with G* = (G′^2^ + G″^2^)^0.5^, and plotted in Figure 18 against the log of frequency ω. The tts can be used to superpose these curves into a mastercurve. We followed Matsumiya and Watanabe’s choice of T_1_ = 115 for the reference mastercurve to check that our values of the shift factors, log a_T_, matched theirs [22]. Table 3 provides those values which were validated by us. Retrospectively, though, the choice of T_1_ = 115 °C for the mastercurve was not the best one since this temperature is right on the transition between ranges mentioned earlier, the (T < T_g_ + 25) range and the ((T_g_ + 25) < T < T_LL_) range (see Figure 25 above).

Figure 19 is the “viscosity mastercurve” at T= 115 °C obtained after shifting horizontally and vertically the data of Figure 18 by an amount log a_T_ and −log a_T_, respectively. The shift factors log a_T_ are given in Table 3. The shift by −log a_T_ on the viscosity axis is due to the definition of the viscosity: η*(ω)= G*/ω, which becomes after superposition: G*/(a_T_ω), so the shifted viscosity using the log scale is: log(G*/ω) −log a_T_.

Our conclusion from Figure 19 is that the tts does not work, at least over the full range of a_T_ω. A closer observation permits to fine tune our conclusion. First, T = 115 °C seems to behave differently than the other 3 frequency sweeps. This is visible at both frequency ends. In the Newtonian region (the plateau region), although it is harder to see without zooming in, the “T115” (T = 115 °C) is the only curve not really merging with the rest (see later in Figure 25 for a more convincing perspective). Second, the overlapping of the 3 frequency sweeps, other than T115, is restricted to a range of frequency that extends from the Newtonian region to the inflective point of the shear-thinning drop off line (at which log η*(ω)~6.46). This restricted range is the only one where we can ascertain that the tts is validated.

Figure 20 plots the temperature variation of log a_T_, the horizontal shift factor. As expected, a_T_ is equal to the ratio of the Newtonian viscosity at T and T_1_:(8)$$loga_{T}=log\frac{\eta_{o}(T)}{\eta_{o}(T_{1})}$$

The red curve in Figure 20 is calculated from Equation (8) using the values of log(η*_o_(T)) in Equation (4) with T_1_ = 115 °C.

Figure 21 displays the variation of χ vs. log ω with temperature for the PS27 of Matsumiya and Watanabe. χ=(G′/G*)^2^ = cos^2^ δ was introduced in Section 1 (Figure 4). Both the position and the magnitude of the maximum vary with temperature. (G′/G*) is the stored elastic energy, which is expected to increase as T decreases, but we observe the opposite trend: the peak maximum amplitude decreases as T decreases. One may think, for a reason, the fact that the modulus is proportional to G_N_ which increases with T. This explanation cannot stand, however, since (G′/G*) is the ratio of two moduli, which cancels out the vertical correction due to the proportionality of the modulus to G_N_ according to the classical tts. In other words, the usual correction on the vertical axis for the temperature dependence of a modulus, T_1_/T, is not required in Figure 21. In addition, as we mentioned before, the temperature span being small, the T_1_/T correction is negligible. According to the current tts, based on the Rouse molecular background, one should not need a vertical shift factor to superpose (G′/G*)^2^ vs. log ω. Figure 21 visually contradicts such a statement: shifting only horizontally will not superpose the data.

Figure 22 is the mastercurve at T = 115 °C obtained after shifting horizontally the data of Figure 21 by the log a_T_ values that were used to shift the viscosity curves of Figure 18 to obtain the mastercurve of Figure 19. We already said that our log a_T_ values matched the values published by Matsumiya and Watanabe [22] for which these authors claimed that the tts works. We see in Figure 22 that when the elastic component of the viscoelastic modulus is used, the time–temperature superposition fails entirely, even in the restricted frequency range it was validated to superpose viscosity in Figure 19. In other words, we face the same dilemma as for the invalidation of the Rouse formula comparing the value of G_N_ from the viscosity side and the elastic side of G* to determine G_N_. In Figure 22 we have drawn a dash straight line (green) joining the peak maxima that shows a tilt from the expected verticality of such a line if the horizontal shifting of the curves the way the tts works had been successful. In other words, if we want to be able to obtain a true mastercurve by shifting the curves in Figure 21, not only do we need to use a vertical shift b_T_ to address the issue of a peak magnitude which varies with T, but we also need to modify a_T_ on the horizontal shifting axis.

The values of b_T_ were found by plotting log(η*(a_T_ω)) vs. χ(ω), not shown, for which we saw that all the maxima at the 4 temperatures lined up horizontally for log(η*(ω)) = 6.46, the value found for the inflection of log(η*(ωa_T_) vs. log(ωa_T_) in Figure 19; we then determined the value of χ at the maximum of χ(ω) from which we determined b_T_ as the ratio of the χ values found at T and T_1_, with T_1_ = 115 °C. The values found for b_T_ are listed in Table 3 and the variation of log b_T_ with T is found in Figure 24. A couple of remarks regarding the procedure to find b_T_: By plotting log(η*(a_T_ω)) vs. χ(ω), the horizontality of the maxima of χ(ω) with T made it not necessary to find a new a_T_, as suggested by the tilt of the green dash line in Figure 22. The choice for b_T_ to be the ratio of the values of χ at the maximum for T and T_1_ was hinted by the considerations we expressed earlier on the possibility to define G′_N_ and G″_N_ in the section on the myth of the Rouse model. In effect, the Rouse model is not capable to understand the need of b_T_ to superpose the χ(ω) at various T, the way the Rouse equations stand (Equation (1). Yet, if we accept to modify the Rouse equations to have G′_N_ and G″_N_ different (with still G″_N_ = G_N_ = ρRT/M)), then b_T_ can be an affine function of χ.

The mastercurve at T = 115 °C obtained by “double-shiftin_G_″ on both the horizontal and vertical axes by log a_T_ and log b_T_, respectively, is shown in Figure 23. The y-axis scale remains linear in this figure and, therefore, the y coordinate is (χ b_T_). The temperature dependence of b_T_ is in Figure 24.

We confirm in Figure 23 that the data of Figure 21 can be superposed, using two shift factors, a_T_ and b_T_, yet the superposition is only valid in the range of frequency up to the maximum of χ(ω). This successful shifting of the data up to the maximum of χ matches what we observed for the successful shifting of the log(η*(ω)) vs. log(ω) up to the inflection point in the shear-thinning range. We know that the correspondence between the two ranges matches because the b_T_ data were obtained at log(η*(ω)) = 6.46, which was also the value obtained at the inflection of log(η*(ω)) vs. log(ωa_T_) in Figure 19.

Figure 24 provides the temperature variation of log b_T_. It is remarkable that log b_T_ vs. T can be fitted by an hyperbolic function of the Vogel-Fulcher type: A + B/(T − C), with the fitting constants A, B and C determined by non-linear regression: A = −0.19876; B = 4.26732; C = 93.78 °C (r^2^ = 0.9999). The value of C was forced to equal the T_g_ of the M = 27,000 monodispersed PS. A loose regression, without forcing the value of C, provided a value of C = 91.00, B = 5.0138 and A = −0.209. The r^2^ is not improved for the loose regression. Let us consider here that C is truly equal to T_g_ for the variation of log b_T_. For the variation of log a_T_ (which can be expressed from the Vogel-Fulcher equation, Equation (4), where C is designated T_∞_), we have shown in [35] how the value of T_g_ and T_∞_ correlate with the isomeric state of the Dual-conformers and their dynamic free volume to determine the value of the T_LL_ transition of the melt. The value of T_LL_ plays an important dynamic role in the Dual-Phase theory of interactions ([2,19], Chs. I.4, II.7 of [20,34]); for our purpose in this section, let’s just say that T_LL_ determines the upper temperature end of the tts applicability that starts at T_g_ + 23 °C, and the need to find a different set of shift factors log a_T_ and log b_T_ when T > T_LL_ to extrapolate the data correctly on the mastercurve for the data in that T region. In addition, T_LL_ also holds many important functions, for instance the end of the dynamism of the dual-phase dissipative statistics, (Ch. 3 of Ref. [19]).

#### Conclusion Regarding the Myth of the Time–Temperature Superposition Principle

The classical claim, e.g., by J.D. Ferry [34], that the Time–Temperature Superposition (tts) can be applied from T = T_g_ to T = T_g_+100 °C to obtain the behavior over the full range of frequency or relaxation times by data shifting extrapolation, is perhaps true for certain polymers under certain circumstances, but we miss the original data to be able to validate the generality of that claim. What we know for certain is that many limitations and restrictions to the general sst must be added to establish it as a workable general rule and that these restrictions are as fundamental or even more fundamental than the sst principle itself to understand the behavior of polymer melts. The restrictions imply that the sst should only be applied over delimited temperature and frequency (time) intervals which depend on the chemical nature of the polymer and its thermal-mechanical history (its processing and thermal history). We have used the specific example of the data of Matsumiya and Watanabe on a classical polystyrene sample to prove the need for certain restrictions that, we claim, should be the ones to be generalized. Here are the specific reservations concerning tts:-The time–temperature superposition principle is not verified for the data we analyzed. Matsumiya and Watanabe recognized in their paper the shortcomings of the superposition applied to G′(ω) and G″(ω), yet they did not question why their data showed such a flagrant discrepancy. We believe that questioning why the tts does not work when performing super standard dynamic rheological experiments on a super standard polymer (PS) was worth the subsequent dedicated analysis time and effort it demands and triggers.-Why do the rheological curves for a simple PS studied in the linear range of visco-elasticity fail to superpose over a classical range of frequency (0.1 to 100 rad/s) using a span of temperature of only 25 °C? Why do the users of the current paradigm of polymer science avoid reporting the failures of a full superposition of their data? Why is there the need to restrict the frequency range or the temperature range for the sst to work? Is there no fundamental requirement for the prevailing theory of viscoelasticity to answer the following questions:
-Why is the tts valid only for the low (left) frequency side of the peak of χ vs. log ω?-Why does log η*(ω) only needs one horizontal shift of the curves, log a_T_, whereas the χ vs. log ω requires two shift factors, log a_T_ and log b_T_ when applying the tts,-Why are we systematically correcting vertically the rheological moduli by (ρT)^−1^ without checking if the Rouse modulus G_N_ does normalize both G′(ω) and G″(ω)?-Why is the value of χ at the maximum increased and not decreased as T increases? This appears counter-intuitive with the explanation that glassy relaxation components are causing the maximum in the χ vs. log ω curve.-Why is the rheological behavior at T = 115 °C different than at T = 120 to 140 °C?

Figure 25 is another way to plot the data to make appear the T_g_ + 25 transition introduced earlier, explain that T = 115 °C is located at the T_g_ + 25 and thus belongs to a different rheological range, with its own shift factor characteristics.

Figure 25 is a graph of ω/χ plotted against log G* as the frequency decreases from left to right in the down sweep procedure used by Matsumiya and Watanabe [22]. As we explain in Ch. 5.4 of Ref. [2], ω′ = ω/χ is the frequency of the elastic dissipative wave that maintains the collective coherence of the melt despite of the local density fluctuation due to the dual-phase interactions. The figure shows that the ω/χ values of the frequency sweeps at T = 140, 130 and 120 °C fuse and overlap at low ω, i.e., at lower values of G*, merging into a single curve like data do in a mastercurve. This is not the case for T = 115 °C which is singled out by showing a minimum and the curve starting to rise sharply at lower ω. This distinct behavior separates out the two regions of viscoelastic across the T_g_ + 23 transition. The presence of one of the isotherms very close to a transition made it impossible to consolidate the tts curves into a mastercurve for this narrow range of temperature interval explored by Matsumiya and Watanabe.

In conclusion, the myth of the time–temperature superposition is linked to the myth of the Rouse model which, we suggested, is in no way descriptive of the rheology of unentangled polymers. The use of the Rouse molecular model as the theoretical base to apply the tts creates huge confusion on the precise way to superpose the data, single or double-shifting, on what range of temperature and frequency, with what correction and depending on which rheological function. In addition, even when the limitations to the superposition are noted, the reason for the restrictions remain obscure and without explanation. As we will see in the next section, the same clueless response to basic fundamental results faces the reptation theory. The mathematical solutions proposed by the reptationist school follow the steps of the Rouse molecular dynamic model to focus on modeling the variations of the chain dimensions during deformation, which, as we have suggested, is the wrong statistical system to model. This fundamental assumption that the dimensions of a single chain are correlated to the macroscopic stresses can be tested experimentally using the Rheo-SANS technique (defined below). As a matter of fact, despite the mathematical brilliance of the reptation work, some recent experimental results fail to agree with the predictions of the reptation theory. This is presented in our next section.

### 2.3. The Great Myth of Reptation. The Failure of the Reptation Model to Correctly Describe and Understand the Shear-Thinning Behavior of Entangled Polymeric Melts (M > M_c_)

#### 2.3.1. The Brilliance of the de Gennes’s Reptation Ideas

The Rouse model was created to describe the viscoelastic behavior of polymer solutions, not polymer melts. The application of the Rouse model to unentangled polymer melts was the initiative of J.D Ferry [32]. It was clear immediately to polymer scientists that the Rouse model could not predict the distinct rheological behavior of entangled melts (or entangled solutions). However, the natural tendency is to start from what is known and to modify it, i.e., in the case of de Gennes, to keep certain basic assumptions of the Rouse formalism while adapting it to the case of reptiles moving within fixed obstacles, which is the title given by de Gennes when he published his first paper in 1971 [8]. De Gennes, who was not a polymer scientist by training, learned from the context of the thoughts on viscoelasticity established at the time. The theory of viscoelasticity of polymers considered then, which still serves as the ground foundation for the current paradigm describing viscoelastic interactions, assumed that the rheological deformation of polymer melts resulted from the behavior of singular chains embedded in a sea of interactions with other chains. In the existing theories of macromolecular physics, the emphasis is placed on determining the shape of the individual macromolecules, often called their chain conformation. The presence of neighboring and interpenetrating macromolecules is perceived as a disturbance to the ideal conformation of the chain. In the traditional texts, the field of interaction responsible for the disturbance is homogeneous. Therefore, de Gennes, like all his predecessors before him, considered the behavior of the melt as the consequences of what happens to a single chain after the effect of the interactions between the chains had been established. De Gennes had the idea of considering the interactions between the chains as a field of obstacles between which a single chain is oscillating through, the way reptiles move, when the chain is requested to move pursuant to an external deformation. De Gennes modeled the motion of the chain among the obstacles using the molecular dynamic language already established in the Rouse model, thus defining the reptation time of a single chain.

In the case of shear deformation, the Newtonian viscosity is classically considered to describe the local internal friction between the bonds of interacting macromolecules which assume a stable thermodynamics state, the equilibrium state at a given temperature and pressure. The non-Newtonian behavior, shear-thinning, is due to a modification by the flow of the dimensions of the macromolecules, i.e., of their conformation, which can be calculated from the effect of the shear rate on the rms end-to-end distance of the macromolecule and the amount of slippage (relax/retraction) occurring. Theoretical models predict that for a shear rate strong enough to overpower the ability of the chain to relax—and this happens at the reptation time—shear-thinning starts to be observed, corresponding to an increase in the rms end-to-end distance of the chain, leading to its orientation. In the classical formulas that describe the non-Newtonian dependence of viscosity with shear rate, the amount of shear-thinning is only a function of two parameters (in addition to the strain rate, of course): the Newtonian viscosity and the value of the reptation time. But these two parameters can be correlated to each other, as in the Rouse’s formula , Equation (3), and to the dimensions and interactions between the chains, which simplifies the description of the flow deformation process to the description of the dependence of the reptation time with temperature, pressure, and chain length (the interactions between the macromolecules, defined by “their entanglement”, is already incorporated in the definition of the reptation time).

In summary, the effect of strain rate, temperature, molecular weight, according to the accepted reptation model, could all be related to a simple explanation: the deformation and relaxation of single macromolecular chains confined to move within the boundaries of a tube, the entanglement tube, whose lifetime was the reptation time. The whole process would continuously be happening, from very low strain rate to high shear-thinning producing strain rate. Additionally, the reptation model provided a new understanding of “entanglement” by quantifying the dimensions of the tube and correlating it to the reptation time. The interactions between the macromolecules could be described topologically, the tube serving as the new topological description of the environment of the bonds.

This was the brilliance, even the beauty, of the original reptation model of de Gennes [2], who succeeded in scaling the effect of all variables into the description of a single parameter, the reptation time. However, this extraordinary tour de force had to be refined over the years to account for a better description of the experimental data, in particular to improve the molecular weight dependence of the reptation time which did not follow the predicted M^3^ variation by de Gennes [3]. The model of reptation in a tube has been significantly improved over the years, by incorporating additional molecular mechanisms such as contour length fluctuation [36,37,38], constraint release [9,39,40,41,42], and chain stretching [43,44,45]. Doi and Edwards [46] proposed to account for the nonlinear rheological behavior by asserting that the external deformation acted on the tube, instead of the polymer chain [47]. The non-affine evolution of chain conformation beyond the Rouse time would be caused by chain retraction within the affinely deformed tube. Other essential improvements to the tube reptation model were done by many contributors, notably Marrucci [9,10], Wagner [11], McLeish [12] but the state-of-the-art version of the tube theory is the GLaMM model (named after Graham, Likhtman, Milner, and McLeish) as it incorporates the effects of reptation, convective constraint release and chain stretch on the microscopic level [45].

It is fair to recognize that the tube model revolutionized the field of polymer dynamics, and stands at present on the highest step of the podium of the current paradigm for its predictions of the linear and nonlinear viscoelastic properties of entangled polymers.

Small-angle neutron scattering (SANS) studies on polymer melts under steady-state flow provide in situ information at a molecular scale on how the flow field is transmitted to the melt. Such experiments, called “Rheo-SANS”, are difficult to set up and require special equipment but their results are fundamental to test experimentally the accepted claim by the reptation model [6,46] that the shear-thinning of entangled polymer chains is due to significant orientation of the segments between entanglements under the shear flow. We quote below two significant Rheo-SANS studies, one by Watanabe et al. in Japan, published in 2007 [48], and the other one by Noirez et al. in France, published in 2009 [49].

Both studies concluded that the chains remained largely un-deformed under steady-state shear flow conditions for which extensive shear-thinning was present. These results represent a formidable challenge to the reptation model of melt deformation [9,36,37,38,39,40,41,42,43,44,45,46,47].

Recently, in 2017, there was the new Rheo-SANS evidence published by Zhe Wang et al. [50], that demonstrates experimentally that the chain retraction step of the tube model does not occur, which led these authors to conclude that our current understanding of the flow and relaxation of entangled polymers, based on the reptation theoretical model of motions pioneered by de Gennes (1971) and Doi-Edwards (1979) is fundamentally wrong:


*“…This result calls for a fundamental revision of the current theoretical picture for nonlinear rheological behavior of entangled polymeric liquids…the predictions by the tube model are not experimentally observed in a well-entangled polystyrene melt after a large uniaxial step deformation”.*


#### 2.3.2. Invalidation of Reptation by Rheo-SANS Results of Watanabe et al. (2007)

In order to examine the chain conformation changes under shear flow for a well-characterized monodispersed entangled polymer and the orientation distribution along the chain backbone, Watanabe et al. examined the Rheo-SANS behavior for an entangled polybutadiene sample dissolved in a deuterated oligomeric butadiene at the volume fraction of 0.28. The rheometer was a Couette apparatus, allowing high-flow shear rates at constant temperature [48]. The shear rate, normalized by the reptation time, was between 24 and 29 s^−1^and at these shear rates the viscosity of the systems was significantly smaller than the zero-shear viscosity (by a factor of ~40). Despite this intense shear-thinning, Watanabe et al. observed that “*the I(q) data just moderately deviate from the Debye function (describing the data at equilibrium)… These SANS data allow us to examine the current molecular picture for the entangled chains under fast shear flow. This picture assumes that successive entanglement segments are not orientationally correlated and behave as independent stress sustaining units even under fast flow… Thus, the above assumption fails for the entangled chains under fast flow".*

In other words, at a shear rate that reduced the Newtonian viscosity by a factor of 40, i.e., under strong non-Newtonian conditions, the chain rms end-to-end distance hardly varied from its value under static (equilibrium) conditions: this result, if verified, was in full contradiction with the basic assumption of the reptation model regarding the deformation mechanism involving the singular macromolecules. Yet, this catastrophic contradiction was kept buried in the archives and was not brought forward by the authors; it remained an isolated research report which was not confirmed.

#### 2.3.3. Invalidation of Reptation by Rheo-SANS Results of Noirez et al. (2009)

Noirez et al., apparently unaware of the results of Watanabe et al. [48], probably for the reasons evoked above, used a similar Quartz Couette rheometer set up and reported on in situ observations of polymer melts under steady-state shear flow using neutron scattering [49]. The amorphous melts studied by these authors were an entangled polybutadiene (T_g_ = −110 °C, M_w_ = 29 M_e_) characterized by a reptation time τ_d_ = 7 × 10^−3^ s (ω_x_ = 143 rad/s) and a low-molecular-weight (unentangled) polybutylacrylate (T_g_ = −64 °C, M_w_~M_e_), characterized by τ_d_ = 10^−3^ s (ω_x_ = 1000 rad/s). Both melts were monodispersed and sheared at room temperature (i.e., far above their respective T_g_). The melts were sheared with a range of strain rates spanning the zone from far below the reptation time to far above it (from 0.011 s^−1^ to 1000 s^−1^) to determine the variation of the chain dimensions across the reptation time and test the admitted reptation theories claims regarding the onset of shear-thinning and of chain orientation/disentanglement [6,46].

Figure 1 of Noirez et al. [49] clearly demonstrates that the two components, azimuthal and longitudinal, of the radius of gyration (R_v_ and R_z_) remained constant at 80 Ả as the shear rate varied from the Newtonian range to a highly shear-thinned melt, and, besides, that no change of the radius of gyration occurred as the melt crossed τ_d_. The authors concluded “*that the chains remain largely undeformed under steady-state shear flow... These observations are of prime importance; they reveal that the flow mechanism and its viscoelastic signature reflect a collective effect and not properties of individual chains".*

We emphasize the last sentence in the conclusion: “…the viscoelastic signature reflects a collective effect and not properties of individuals chains”. This is the key sentence to remember from this experimental research. In summary, both Watanabe et al. and Noirez et al. concluded that the macromolecular dimensions remain quasi-unchanged as the melt is sheared in the non-Newtonian region, and this conflicts totally with the currently accepted understanding of shear-thinning. The failure of the existing models to interpret such a fundamental aspect of polymer rheology cannot remain unchallenged [13,14,15,16,17,18,19,20,21].

#### 2.3.4. Invalidation of Reptation by Rheo-SANS Evidence That Chain Retraction Does Not Occur by Zhe Wang et al., 2017

This paper by Zhe Wang and 11 other co-authors [50] solves the problem of critically testing the chain retraction hypothesis of the tube theory for entangled polymers. In principle, these authors explain in their paper, one should be able to critically test the chain retraction hypothesis by performing SANS experiments on uniaxially stretched entangled polymer melts and comparing the measured R_g_ with theoretical predictions. “*In reality, experimentalists have encountered tremendous difficulty in following this approach…it is practically impossible to reliably determine the radius of gyration tensor through model independent Guinier analysis, because of the limited Q range and flux of existing SANS instruments and the large molecular size of entangled polymers*”. These limitations of the analysis of the radius of gyration tensor in step-strain relaxation Rheo-SANS investigations may represent arguments to question the results of Noirez et al. [49] or Watanabe et al. [48] above.

Zhe Wang et al. recently recognized the value of “spherical harmonic expansion” as a general approach for characterizing Q-dependent deformation anisotropy and chain conformation at different length scales. The idea of using spherical harmonic expansion of the orientation distribution function of statistical segments in deformed polymer networks was conceived by Roe and Kribaum who discussed the potential application of this technique to analyze the amorphous halo for stretched polymers [50]. A more formal treatment of the measured scattering intensity by Legendre expansion was developed by Mitchell [Refs. 84–86 of [50]] and applied to the tensile deformation mode. The originality of Zhe Wang et al.’s work is to have applied the spherical expansion analysis to test directly and unambiguously the chain retraction hypothesis, central to the theoretical picture of the tube model.

The stretching of the rectangular samples of PS to orient them before their SANS analysis was conducted by Zhe Wang et al. by uniaxial elongation at 130 °C to a stretch ratio λ = 1.8, with a constant crosshead velocity v = 40 l_0_ /τ_R_, where l_0_ is the initial length of the sample, and τ_R_ the Rouse relaxation time (~600 s). The oriented samples were allowed to relax for different amounts of time (from 0 to 20τ_R_) at 130 °C and then were immediately quenched by pumping cold air into the oven. The authors verified that they successfully froze the conformation of the polymer chain with negligible stress relaxation during the quenching procedure.

Zhe Wang et al unambiguously showed that:


*“the two prominent spectral features associated with the chain retraction—peak shift of the leading anisotropic spherical harmonic expansion coefficient and anisotropy inversion in the intermediate wave number (Q) range around Rouse time—were not experimentally observed in a well-entangled polystyrene melt after a large uniaxial step deformation”.*


They added:


*“Unlike the previous investigations, there is no ambiguity associated with model fitting and no room for human bias. Therefore, our critical test clearly demonstrates that the chain retraction hypothesis of the tube model is not supported by small-angle neutron scattering experiments.”*



**
*“This result calls for a fundamental revision of the current theoretical picture for nonlinear rheological behavior of entangled polymeric liquids.”*
**



*“Therefore, without an alternative mechanism for molecular relaxation, the idea of nonaffine deformation alone does not seem to be able to explain the experimental observation.”*



*“Since the tube theory is of paramount importance for our current understanding of the flow and deformation behavior of entangled polymers, the invalidation of the chain retraction hypothesis has immense ramifications.”*


#### 2.3.5. Conclusion on the Great Myth of the Applicability of the Reptation Model to Entangled Polymer Melts (M > Mc)

Despite all its elegance, mathematical sophistication and quasi-general acceptation we conclude that the reptation model incorrectly describes the Rheo-SANS experiments of Watanebe et al. [48], Noirez et al. [49], and of Zhe Wang et al. [50] and **should be abandoned**. The reason for this radical proposition, in our view, is that the dynamics of the interactions defining the melt properties should not be defined by statistical systems which are the single macromolecules. The failure of the reptation model also implies re-considering the concept of entanglement, the corner stone of polymer physics which, in our opinion, is not understood by the current paradigm of polymer physics.

### 2.4. Shear-Refinement and Sustained Orientation: The Lack of Understanding by the Current Paradigm

#### 2.4.1. Shear-Refinement

“Shear-Refinement” is the observed influence on subsequent viscoelastic behavior (e.g., viscosity) of a pre-shearing treatment of a polymeric melt. Cogswell mentions the influence of thermo-mechanical history on viscosity in his book [51], p. 53:


*“Intense working, producing high shear, will usually lead to a reduction in viscosity and also a decrease in the elastic response”.*


Note that the viscosity reduction discussed by Cogswell is not due to a decrease in molecular weight, which is known to occur concomitantly, to a variable degree depending on the polymer and the experimental processing conditions.

##### Pre-Treatment on Branched Polymers

Most of the pioneering work was conducted 20 years ago on branched polymers (PE,PP) by such authors as D. E. Hanson [52], M. Rokudai [53], B. Maxwell [54], J.-F. Agassant [55], H. P. Schreiber [56] (who wrote a review of the subject up to 1966), G. Ritzau [57,58], who provides details of a shear-refinement apparatus, J.R. Leblans and Bastiaansen [59], Van Prooyen et al [60], Munstedt [61], who studied the effect of thermal elongational history, and A. Ram and L. Izailov [62]. Hanson [52] showed that the Melt Flow Index of a branched PE could be modified by shear-refinement from 0.28 to 0.66 and that the MFI returned to the initial value 0.27 after solution and re-precipitation of the pre-sheared sample. Cogswell [51] comments as follows on the results obtained by Hanson and others [52,53,54]:


*“The change is seen to be reversible by solution treatment. Molecular weight characterization indicated that all these samples were identical… [Shear-refinement effects] “might at first appear to be the result of degrading the polymer, are frequently reversed by cooking the melt, though the time for which the melt may need to be cooked to achieve reversion may be much longer than the natural time of the material (viscosity/modulus at zero shear)”.*


J.-F. Agassant et al. [55] show that the effects of shear-refinement are most obvious, and most commonly exploited, in the case of PVC which is known to have a morphology very sensitive to thermo-mechanical history.

No clear explanation was ever given to the origins of shear-refinement by these authors, which remained empirical until Bourrigaud [39] published a possible reptation-based interpretation in the case of branched polymers.

Bourrigaud [63], and Berger [64] have recently investigated the shear-refinement of long-chain branched (“LCB”) polyolefins in their thesis. Bourrigaud focused on several well-characterized low-density branched polyethylene grades and obtained proof of the influence of the strain amplitude of shear deformation on the degree of viscosity reduction during subsequent processing. Bourrigaud suggested that molecular topology is critical, and his results support the view that molecules with very long-chain branches are highly affected by shear refinement, whereas linear polyethylene seems to undergo much smaller changes (if any), under the experimental shear refinement conditions he used. Bourrigaud and co-workers [65] concluded that the degree of long-chain branching or ramification qualifies or disqualifies, for the most part, the degree of viscosity reduction observed by shear refinement. In other words, controlled alteration by branching of the molecular weight distribution leads to the optimization of shear-refinement and of its benefits, according to these authors. Furthermore, Bourrigaud et al. showed that refinement by elongation is more effective than refinement by shear for the same flow strength [63,65]. Berger [64] and Berger et al. [66], worked with a long-chain-branched polypropylene under very high shear strain rates and found similar results. Additionally, Berger and coworkers [66] confirmed that the MFI of branched PP, collected as pellets, could be increased by shear-refinement, and that solvent dissolution would reverse the effect; after evaporation of the solvent, the MFI returning to its original value. These authors concluded that disentanglement was responsible for the decrease of viscosity and die swell [66]:


*The pre-treatment of the LCB-PP in the capillary rheometer at the highest shear stress applied causes a significant reduction of the tensile stress, which can be referred to the reduction of the mass-average molar mass. However, the significant decrease of the extrudate swell after the pre-treatment cannot be explained by the change of the molar mass, as the elastic behavior of polymer melts is known to be independent of the mass-average molar mass. Therefore, the reduction of the extrudate swell is an indication of a change of the entanglement network during the pre-treatment.*


##### Pre-Treatment on Linear Polymers

We published a series of papers and patents during the last two decades [67,68,69] to explain how the combination of shear rate and controlled strain mechanical treatments applied prior to or during processing of linear polymers (not branched) could result in substantial viscosity reduction benefits that allow, for instance, to work in extrusion at lower temperatures or under lower pressures at the same throughput. We invented, designed and ran “Rheo-Fluidizers”, processing equipment making use of vibrational methods during melt extrusion to induce shear-refinement by shear strain energy coupled with extensional flow [67,68,69]. The emphasis of this “dynamic shear strain refinement” process was on the improved processability of **linear** high-molecular-weight polymer melts, such as polycarbonate and Plexiglas (PMMA), i.e., polymers without branches. We showed [68,69] that to induce the shear refinement benefits, a combination of shear stress and superposed oscillation could raise the elasticity of the melt to a level identical or perhaps even superior to what branching could do. In other words, we proposed that, at least under dynamic conditions, both linear polymers and branched polymers could qualify for “disentanglement” by shear strain refinement. Furthermore, we drew attention to the requirement of rheological criteria to be fulfilled for shear refinement to occur [Ref. [2] pp. 100, 110, 229], and pointed out the importance of the shear strain amplitude of the oscillation to operate the melt in the non-linear time-dependent viscoelastic range. We suggested that the combination of shear-thinning and strain softening during the pre-treatment, which we designated “Rheo-Fluidification”, could produce either shear-refinement benefits [Ref. [2] Section 4.6], or Sustained-Orientation (“disentanglement”) depending on certain conditions [67,68,69]. Sustained-Orientation is explained in the next Section 2.4.2.

Shear-refinement work has remained largely empirical because of the lack of its understanding by the current models. The viscosity reduction is temporary and rheological properties can be restored, which can occur in various ways and was not very well understood until now. Most of the comprehension necessary for its generalization and extrapolation to all macromolecules was lacking because the current models remained speechless about the shear-refinement results. For instance, for linear polymers, the current paradigm could not understand how it could be possible that shear-refinement effects could happen since the chains were linear and not branched. In addition, the Rheo-Fluidified melts had relaxation times calculated from their cross-over frequency much shorter than those with the same molecular weight without treatment, and this was as if they had been “disentangled”, sometimes by a factor of 1000 or even 10 times that. The claim by Munstedt [70] that shear-refinement can only exist for branched polymer structures and not for linear chains is debated in Section 2.4.3 below and in [15].

It is clear that the lack of comprehension of shear-refinement for linear polymers by the current models poses a threat to the whole foundation of the existing paradigm in polymer science. The situation is different for branched polymers for which Bourrigaud’s theoretical explanation has the merit to search for a classical interpretation [63]. Bourrigaud modified the McLeish and Larson’s pom-pom model [12] to account for the increase, due to branching, in value of the tube renewal relaxation time and explained, at least partially, some of the shear-refinement results for its branched PE samples. For linear polymers, however, “disentangled” polymers present a real challenge to existing models of flow.

The positioning of the science community with respect to the “disentanglement” results remains confused and hesitant based on the claim of the reptationists’ gate keepers that those results must be artifacts since they disprove their theory. Properties of melts brought out of equilibrium are largely ignored. Yet, many plastic industries are directly concerned and will benefit from the fundamental understanding of what causes shear refinement viscosity drops in linear or branched polymer, and how this can be applied to processing of polymer resins, branched or not. The ability to process plastic melt at much lower temperatures (50–80 °C below normal), because of reduced viscosity due to shear-refinement or disentanglement, opens up new boundaries not just in processing but also in blending, such as in nanoparticle dispersion, or for the processing of high-temperature sensitive additives (wood flour, instable additives such as peroxides, etc.). Details are given elsewhere [Ch. 8 of Ref. [2]].

#### 2.4.2. Sustained Orientation

Shear-refinement can occur with unentangled polymers, linear or branched, and therefore shear-refinement should not always be called “disentanglement”, like we did in our early publications (when we were not even aware of the work of others on shear-refinement). It is true that we only applied our Rheo-Fluidification pre-treatments on entangled melts, because of the commercial applications of reducing their viscosity, and this was one of the reasons to designate the results “disentanglement”. When entangled polymer melts are submitted to Rheo-Fluidification treatments, the result produced is at least shear-refinement, at best Sustained-Orientation, and the distinction means that the Sustained-Orientation is more difficult to achieve, requiring a dual-phase model understanding of the differences between unentangled and entangled melts, in particular their stability under stress.

In simple terms, by manipulation of the stability of entanglements, it is possible to create and maintain quasi-stable at high temperatures in an amorphous polymeric melt (say 120 °C above T_g_) a certain state of orientation that was induced by a mechanical deformation. The manipulation of entanglements was achieved by coupling two Rheo-Fluidification processors Section 2.2, Figures 2.1–2.4 in Ref. [2]. The “sustained-orientation” discovery describes the possibility to obtain non-equilibrium entanglement states for polymeric melts which can be preserved in a pellet formed after the treatment. This pellet displays a melt flow index (MFI) that can be 100% larger than the original (virgin) pellet before the treatment, after correction for any molecular degradation present due to the process. This new state of polymer matter challenges the current established models of polymer physics, because such “oriented” melts can remain oriented for hours at temperatures below their T_LL_ transition temperature, yet can slowly recover in time their initial un-oriented equilibrium state (the MFI of the treated pellet then slowly reverses to its original MFI). This esoteric behavior can be understood by the Dual-Phase model of the interactions that explains entanglements as a split of the statistical system of interactions yielding a set of cross-dual-phases [20].

The experiments of “sustained-orientation” could be interpreted qualitatively using the classical terminology by a change in M_e_, the molecular weight between entanglement (thus the wording which was used, “disentanglement” or “re-entanglement”), except that there is no classical explanation to why M_e_ could vary so slowly in time, M_e_(t), independently of the terminal relaxation time, and be increased or decreased by relatively low shear forces. For instance, using the classical language, sustained-orientation would produce a melt with an M_e_ value twice as big as the virgin pellet, M_eo_. That value can be frozen in the new pellet and stay stable as the pellet is reheated above the T_g_, say at T = T_g_ + 120 °C, at least for a certain time that could be equal to a million times the value of the reptation time. The M_e_(t) can then start to decrease towards its original equilibrium value M_eo_, the time to control the return to equilibrium being controlled by pressure. There is no explanation in the current theories for an unstable entanglement network resulting in an unstable liquid state for polymers, and on how the instability dynamics could be correlated to non-linear viscoelastic effects.

What sustained-orientation suggests is that the classical concept of M_e_ to describe entanglements is overly simplistic and its usefulness is, at best, limited to the linear range of viscoelasticity. The whole foundation of polymer physics, based on its understanding of entanglements, appears to be challenged, perhaps even overhauled, by the type of experimental results resulting in Sustained-Orientation.

Figure 26 below (similar to Figures 4–9 for PC and Figure 4.74 for PMMA in Ref. [2]) demonstrates the benefits of Sustained-Orientation, sometimes designated “disentanglement in a pellet” in contrast to “disentanglement in-line” which refers to the shear-refinement reductions in viscosity and melt elasticity while the melt is being processed after the pre-treatment.

Figure 26 applies to a linear PC grade. One compares the MFI value found for pellets made from a melt prepared by the twin Rheo-Fluidification treatment stations of Figure 5a,b of Ref. [2] with the value of viscosity measured by the in-line viscometer (shown in Figure 4.8 of [2]) at the exit of the strand die. Although the two temperatures are different (300 °C for the MFI measurement, 275 °C for the in-line measurement), the correlation is validated: when the in-line viscosity drops, the pellet has a higher MFI than the reference pellet (11.3). In other words, the viscosity benefits obtained from the manipulation of the melt stability can be frozen into a state in a pellet that will survive subsequent heating periods, about 20,000 times its terminal relaxation time value at 150 °C above its T_g_. This “Sustained-Orientation” behavior shambles completely the current understanding of viscoelasticity in polymer melts.

Ever since we were able to produce hundreds of pounds of linear polymers (PC, PMMA, LLDPE) exhibiting the Sustained-Orientation behavior and understood that this new property contradicted the current paradigm of polymer physics, we knew that a different explanation of “entanglements” was required and that polymer physics had to be reconstructed from a different understanding of the coupling between the covalent and inter-molecular interactions.

We conclude this section by claiming that the reptation tube model, as it stands now, cannot explain the challenging results obtained by “shear-refinement”, and by “Sustained-Orientation”. If a model cannot comprehend a phenomenon that we can reproduce to produce batches of hundreds of pounds of pellets demonstrating the benefits of the phenomenon at a time, then this model should be abandoned. This is the way science works.

#### 2.4.3. The Munstedt’s Exclusive Requisite That the Polymer Must Be an LCB (with Long Chain Branches) to Be Able to Obtain Shear-Refinement

Munstedt [70] recently claimed that only branched polymers could demonstrate shear-refinement benefits, not linear polymers. According to him, linear polymers could only show artifacts or unreported degradation [70]. We offered a rebuttal to Munstedt’s paper and his allegations [15]. These two publications should be read to illustrate how the gate keepers of an existing paradigm practice their censorship power to eliminate any possible existential threat. The rebuttal, for instance, was rejected by the Journal which published the paper by Munstedt (Journal of Rheology). Also, Munstedt misquoted and mischaracterized—intentionally or not is debatable—some parts of Ref. [2] to denigrate the results. Let us stay on course and only concentrate on excerpts from Ref. [15] relevant to the present discussion.

##### Münstedt’s Critical Condition That Branching Must Be Present to Observe Shear-Refinement Is Wrong

In Ref. [2], we introduce new equations to analyze the rheology of melts (shear-thinning, strain-softening) in terms of the Dual-Phase model and show that they also explain the origin of the rheological instability. The long-term retention of the lower viscosity in the Rheo-fluidified pellets when re-heated to a melt state, sometimes for times several hundred thousand times greater than the reptation time at that temperature, represents an immense challenge to the currently admitted models of chain dynamics such as reptation. This challenge is not acknowledged by the community of rheologists, except swept away as artifact, such as in the paper by Münstedt [70]. However, how could this be an artifact when produced several lots of 150 lbs of **sustained-oriented pellets**, the product of the “artifact”, which could regain in time their original viscosity after re-melting!

We concluded in [2] that this “Sustained Orientation” paradox is linked to a new concept: the instability of the Dual-Phase of the interactions. A first degree instability can be induced by a combination of shear-thinning and strain softening that may result in shear-refinement effects. Sustained-Orientation requires certain conditions in addition to the 1st degree instability criteria to trigger an instability of the 2nd kind: the instability of the Cross-Dual-Phase entanglement structure.

There are two types of sources to trigger the rheological instabilities of polymer melts: one is controlled by the recoverable dynamic free volume variations, the other by the modification of the entanglement network structure, by entropic dissipation (orientation of the network). This competition between these 2 mechanisms of instability is different for a given polymer and represents the true debate to have regarding the shear-refinement results, as we emphasized to Münstedt, during our intensive discussions [15]. For instance, the Dual-Phase model of the interactions that we have introduced in Ch. 1 of Ref. [2], a book reviewed by Munstedt, explains the dynamic source of the free volume, which is also influenced by the topology of the chains, in particular whether long chain branching, short chain branching or no branching is present. Both the **amount** and the **structure** of the dynamic free volume are influenced by branching. However, and this is missing in Munstedt’s analysis, they are also influenced by other rheological factors, the orientation of the chains, the frequency and the amplitude of a vibration of the coherent interactive medium, the pressure in the melt, etc., all these parameters influence the local density of the melt and the frequency of the elastic dissipative wave that compensates for the local packing density inhomogeneity. In turn, they also influence the melt modulus (the famous G_N_ = ρ RT/M correlation), and thus influence shear-thinning and strain softening. Münstedt focused on the presence of the long chain branches to determine a criterion for shear-refinement [70]. We argued that to understand why shear-refinement can occur in both branched and linear polymers one needed crucial information that are never provided by the molecular models: 1. the determination of the local packing density and of the localization of the free volume in the structure, and 2. the influence of branching on these two variables. The Dual-Phase model is easily applicable to this situation [2] because of the local cross-duality between the F/b dissipative states and the conformational states (trans ,cis, gauche). This (F/b ↔ (c,g,t)) local cross-duality also predicts the influence of vibration, shear rate and shear strain on the free volume amount and its distribution, in particular how to increase it, whether the basic polymer is branched or linear. Therefore, the topological criteria by Münstedt that branching **must be** present to observe the conditions for shear-refinement is simply wrong.

### 2.5. Strain-Induced Time Dependence of Rheological Functions

The conditions to achieve linear viscoelasticity are obtained at low strain amplitude for dynamic rheological experiments, where an oscillating strain is applied to a molten melt with frequency ω at temperature T. Under such conditions, the elastic and loss moduli, G′(ω,T) and G″(ω,T), respectively, are independent of the value of the strain, that could be 1%, 3%, 5%, etc., up to the limit of linear viscoelasticity. The limit of linear rheology is reached when the value of the moduli become strain dependent, i.e., when the stress is no longer proportional to the strain (non-affine deformation). The determination of this limit is compulsory before running any other tests in the linear region of viscoelasticity; it is done by running a strain sweep at given T and ω. The value of the strain is increased continuously and slowly until a deviation from the horizontality of the modulus appears that marks the beginning of non-linearity. In the following we are interested in the “stability” of the non-linear solution, meaning once we have reached the value of strain for non-linearity are the moduli values stable in time or starting to drift to make them time dependent?

Does a non-linear state obtained by increasing strain become *immediately* instable: time dependency starts as soon as its modulus differs from its linear value?Is the strain value for the end of linear viscoelasticity different from the strain value for the start of the time dependency of the non-linear modulus?Is the rate of the time dependency of modulus a function of the strain?

The general affirmative response for polymer melts answers question #2, adding in complement that the response is function of the chemical nature of the polymer, the value of ω, of (T − T_g_) and of the strain.

In other publications (Ch. I.7 and II.9 of [20]), we address the issue of determining the critical strain at which the instability of the non-linear rheology is triggered (question #2) and the influence of strain on the rate of the moduli decay (question #3).

When we say “instable”, we are **not** talking about a chemical instability of some sort (degradation, esterification, cross-linking) or a surface instability (cracks, edge fracture, surface contact loss) that would alter the measurement, we are talking about the possibility to re-organize the interactions inside the material under stress that results in the time dependency of the moduli. For instance, using the language of the Dual-Phase model, we want to know if the dynamic free volume restructures (i.e., the (b/F ↔ (c,g,t) kinetics evolves), or, for entangled melts, whether there is an enthalpic or entropic modification of the compensation between the two dual-phases, when the strain brings the system in the time-dependent non-linear range (“disentanglement”).

It is clear that we need to ensure that the chemical instabilities or surface instabilities are not responsible for the time dependency observations. This point is crucial to consider in detail (see [71] section 4.4.2 “Challenging Interpretations”: 4.4.2.1 “Viscous Heating. 4.4.2.2 “Shear Degradation. 4.4.2.3 “Drooling of the Melt outside the Rheometer Plates. 4.4.2.4 “Plastification Due to the Monomer Concentration Increase by the Shear Stress. 4.4.2.5 “Shear-Thinning. 4.4.2.6 “Edge Fracture Explanation. 4.4.2.6.1 “Melt Fracture Initiation: Vinogradov’s Criteria. 4.4.2.6.2 “Simultaneous Dielectric and Dynamic Mechasnical Measurements in the Molten State. 4.4.2.6.3 “Effect of the Nature of the Surface Melt Contact”), because certain testing configurations are more inclined to create artifacts than others and we need to cross-reference those doubtful results with results that can be trusted 100%. This dedication to scrutinize each experiment details to eliminate any potential artifact pitfalls is the absolute norm when dealing with the experiments exhibiting a strain induced time dependence of the rheological functions. This takes extra dedicated time [15,71] but is necessary to counter the artifact reflex of the deniers of such results by the gate keepers of the existing paradigm [70].

#### 2.5.1. First Example of Strain Induced Transient of Viscosity (Inducement)

Figure 27a,b concern a PS melt studied with a dynamic rheometer (AR 2000, TA Instruments) in the time-sweep mode. The temperature is 165 °C (65° above the T_g_ of PS) and the frequency remains equal to 20 Hz (ω = 125 rad/s). The cross-over frequency for this PS at that temperature is 0.1 rad/s, so the Rouse time is 10 s. The initial strain is 5%, known to be in the linear viscoelastic range. The strain remains constant to this value for 3 min, then it is automatically increased to a new value, 10%, where it stays constant for 3 min; this action is repeated until the final strain is 23%. In other words, the strain varies step wisely every 3 min from 5% to 23%, the sample undergoing time-sweep steps lasting 3 min between each increase of the strain. Figure 27a displays the viscosity η*(ω) vs. Time for each time sweep for strain equal to: 5%, 10%, 15.2%, 17.5%, 20% and 23%. We record the value of G′ and G″ during each of the 3 min steps. Figure 27b provides the variation of G′(t) and G″(t) for the last step, corresponding to 23% of strain (ω is still 125 rad/s).

It is apparent in Figure 27a that a time dependent (transient) behavior is triggered by the increase of strain at 15.2%. For 5% and 10% strain, the viscosity remains constant, but at 15.2% in Figure 27a, the viscosity starts to decay. The magnitude of the effect increases with strain, the rate of the decay does too (compare the viscosity curves for 15.2% strain (green triangles up) and 17.5% (green triangles down): the increase of the slope is proportional to the rate increase. As the strain increases, the apparent straight line decay becomes an exponential decay visible by the convex curvature. This is particularly visible for the 23% strain time sweep. Note that the decay of the moduli in Figure 27b is not over and has not reached a plateau after 3 min, which contrasts with a terminal relaxation time of 10 s for this melt. The time scale involved in the transient decay is very different from the molecular time scale. There is a classical “engineer” description of this phenomenon in terms of shear-thinning and strain softening: at ω = 125 rad/s T = 165 °C, the melt shear-thins, i.e., its viscosity drops from the Newtonian value at that temperature to a lower value, 1075 Pa-s in Figure 27a (@ 5% strain). The effect of strain on the modulus, a non-linear effect, is called “strain-softening”, which is quantified by the ratio, h, of the non-linear modulus to the linear modulus (h < 1). Shear-thinning is controlled by the value of ω, strain-softening by the value of the strain, γ, so it is expected that the strain rate maximum per cycle, ωγ, play a role to determine the onset of the time dependence behavior which we can designate by either “the instability of strain softening by the frequency ω” or “the instability of shear-thinning by the strain γ”. Criteria of melt instability based on the value of the strain rate and/or the strain have been used to study non-linear effects such as melt fracture or melt flow non laminar decohesion [72,73]. It is important to verify [71] that none of these critical values for melt inhomogeneity is reached to explain the triggering, at such a low γ (15%), of the transient behavior observed in Figure 27a,b.

Wang [73] has established that two criteria must be met simultaneously to trigger a non-laminar structure of the melt in a gap: one of these criteria relates to the strain rate, the other to the strain. The strain criterion of Wang (γ > 100%) is not met, by far, in Figure 27 since the transient occurs for γ = 15.2%. The possibility that melt fracture occurred at the edge of the sample to explain the stress and viscosity decay in Figure 27 has also been considered and contradicted [72]. A simple experimental way to eliminate such an explanation for the strain induced triggering of a rheological transient is to consider if the phenomenon is reversible. This is shown in the next example on another polymer, a linear low density PE.

#### 2.5.2. Second Example of Strain Induced Transient of Viscosity (Inducement and Recovery)

Figure 28a,b summarize schematically the frequency and strain experimental profile to create a transient with a laboratory dynamic rheometer, and demonstrate that the phenomenon is reversible upon cessation of the cause of the effect.

The data were obtained with a dynamic rheometer, the ARES from Rheometrics, using a parallel plate configuration, but a cone and plate combination was also used, providing essentially the same results. The resin was a LLDPE from Dupont-Dow Elastomers (Engage 8180), the temperature was 155 °C, and the gap was chosen between 1.2 and 2 mm.

Figure 28a,b describe the frequency and % strain history. Figure 29 plots dynamic viscosity against time. The first and last segment, called “initial” and “recovery” in Figure 28a,b represent the baseline, the value of viscosity under linear viscoelastic conditions, i.e., under very low frequency and amplitude (here 1 rad/s, 1% strain). The so-called «**treatment zone**» in Figure 28 and Figure 29 was initiated by a jump of the frequency, from 1 to 47 rad/s, which created, in Figure 29, an “instantaneous” drop of viscosity from 57,000 Pa-s to 10,000 Pa-s, due to shear-thinning. The jump was then followed by a gradual stepwise increase of the strain amplitude, from 1% to 25%. Figure 29 shows that for the first 2 steps of increase of strain, the viscosity held constant at 10,000 Pa-s, its shear-thinned value at that temperature and frequency, but that starting at strain = 13%, the viscosity started to become transient declining from 10,000 to a steady state value of 3100 Pa-s. The decay of the viscosity took about 25 min. The frequency and strain amplitude were then changed back to their low values of the linear range (1%, 1 rad/s), and one observes an “instantaneous” partial loss of the effect of shear-thinning combined with strain softening, i.e., the viscosity jumped back to 38,000 Pa-s. Further recovery of viscosity occurred over the following 20 min, viscosity increasing slowly and finally regaining its original Newtonian value, 57,000 Pa-s. In other words, the state of the melt produced by the transient treatment was unstable when the energy that produced the transient behavior was released: this is why viscosity slowly increased in time and returned back to the original value for the melt. Nevertheless, it took 20 min for recovery, and this time is 60 times longer than the terminal time at that temperature, making it possible to exploit the benefits of a smaller viscosity during recovery if the melt were to be processed at that stage. One can define the viscosity benefit by comparing the initial Newtonian viscosity (57,000) and the Newtonian viscosity before recovery after the shear-thinning elastic loss (38,000), a ratio of 1.5 in this treatment **(«50% viscosity drop»).** Notice that a processor could still benefit from shear-thinning of the treated resin (Figure 28), and work under much greater viscosity reduction (3100 Pa-s versus 57,000 Pa-s, an improvement of over 1700% !).

The experimental procedure described in Figure 28 has many variations: the time duration between strain amplitude step-ups can vary, the strain amplitude increment itself can be changed as can the temperature of the melt and the frequency of operation during treatment. The treatment could also be done differently, by increasing at low frequency the strain to 25%, say, and step wisely increasing the frequency from 1 rad/s to 47 rad/s. All these changes contribute to the final % viscosity reduction, which can be as small as 20%, to as large as 3000%. The wrong procedure can also produce artifacts or surface effects, as is explained in [71].

#### 2.5.3. Conclusions on the Strain-Induced Time Dependence of Rheological Variables

##### The “Process Engineer” Interpretation of the Results

Strain softening, known to decrease the modulus at higher strain, combines with shear-thinning due to the effect of frequency to render the melt unstable in its original entanglement network configuration; thus the transient behavior occurs. In a step strain experiment conducted in the molten state, a softening factor is defined, h = G(strain)/G(LVE), where G(strain) is the melt modulus for a given strain and G(LVE) refers to the strain independent Linear Viscoelastic Value (h < 1). At low strain, the modulus is only time dependent, and an increase of strain produces an increase of stress proportionally. Pure viscometry experiments have demonstrated that above certain strain rates, corresponding to a certain stress level, a transient decay towards steady state released the elastic energy stored during initialization. It was suggested in earlier publications dedicated to more engineering audiences [12,67,68,69,70,71,72,73,74,75,76] that the dynamics of this process could be viewed as a recursive effect of the stress on relaxation times. As stress continues to grow, due to increased strain, strain softening is the first revealing sign of the modification of the structure due to the stress dependence of the relaxation time. Figure 29 reveals that under dynamic conditions, the softening factor h can become time dependent, which translates into a transient behavior. The advantage of producing transient behavior with a dynamic viscometer is that G′ and G″ become time dependent, so it is possible to analyze these curves individually and also follow how (G′/G*)^2^ varies during transient stress decay. The transient decay can be produced in-situ in the rheometer, and a frequency sweep performed before the transient and after it, allowing an easy way to analyze the differences due to the stay in the non-linear regime. This type of experiments allows us to analyze the influence of strain and frequency during time sweep (“the treatment”). Additionally, the fact that a frequency sweep in the linear regime can be performed on the sample after it has been treated non-linearly, proves the integrity of the sample and its surfaces, in contradiction to the claims by Munstedt [70] that the treatment conditions degraded the sample integrity.

##### The Theoretical Physicist Interpretation of the Results

We have studied many curves like those in Figure 27 and Figure 29, obtained using a parallel plate configuration, a cone and plate and a Couette configuration, using many different polymers, using different temperatures, different molecular weights, under pressure in a confined environment with no edges, superposed to extrusion flow, under cross-lateral vibration etc.(Ref. [2] is dedicated to report in details those experiments and results), and the same conclusion imposes itself: the rheological phenomenon observed that is triggered by strain has a kinetic origin which makes it vary with frequency and temperature but **does not work at the same scale as the terminal time**, τ_p_ = 1/ω_x_: it refers to a different phenomenon that is not accounted for in any previous model of viscoelasticity: the dissipative aspect of the interactions. In our theoretical work on the Grain-Field Statistics of open dissipative systems [21], this concept is embedded in the equations regulating the interactions between the dual-conformers, and these assumptions are applied to polymers in [19]. The “dissipative aspect” means, in essence, that beyond enthalpic and entropic changes occurring to constrained systems brought out of equilibrium, the size of the systems may restructure, rendering the statistical frame definition to become part of the dynamics. This fundamentally different statistical approach is what fuels the new paradigm of the interactions that we introduce which, in many ways, explains the shortcomings of the current paradigm to be able to correctly address the experimental results presented in this paper. One could say, to simplify, that the new paradigm fuses with the current paradigm, which may then regain some merit, when the system of interactions is in a state above the T_LL_ transition, a typical “dissipative transition” resulting from the dissipative nature of the interactions [19,24,34].

## 3. Conclusions

The deformation of a polymer melt in shear mode represents the main subject of interest in the science of rheology of such materials. It is a crucial topic for successfully processing these materials. In the above examples that dealt with linear viscoelastic rheological conditions with no effect of strain, in Section 1, Section 2 and Section 3, we saw that even in these simple conditions the Rouse model failed to satisfactorily describe the data of unentangled melts when carefully comparing experiments and theoretical predictions. The same failure of the reptation model was also demonstrated when comparing the calculated projections of the affine and non-affine hypotheses suggested by the reptation model of entangled melts with the experimental results obtained by Rheo-SANS. In summary, even in the linear range of viscoelasticity the acclaimed Rouse and de Gennes models are challenged by experimental evidence. In the non-linear range, at a high strain rate and strain, the subject of the other examples presented in this paper (Section 2.4 and Section 2.5), it is generally admitted that the current theoretical developments that successfully predict the main characteristics of polymer melts in the linear range fall short, but merely need improving and tweaking of the parameters. The extrapolation to the non-linear behavior generally consist of adding some terms to the mathematical formulation of the linear viscoelastic model. As we stated at the beginning of this paper, all the current models in polymer physics are based on “chain dynamics statistics” [6,7,8,9,10,11,12]. The aura these polymer dynamic models have reached among the polymer scientific community makes them the current standard references that control the field of plastic engineering that relies on the understanding of viscoelasticity and rubber elasticity. Yet, as we suggest, it is possible that the experiments described in this work challenge the current paradigm to its limits, to the edge of its usefulness.

The present understanding of the physics of macromolecules is based on an analysis of the properties of a single chain. The presence of the other chains is perceived as a mean field influence on the properties of that chain. The reptation school considers that this mean field can be looked at as a topology, a homogeneous field of obstacles restricting the motion of the single chain, which is claimed to explain the extra molecular weight dependence of viscosity at M_c_ and beyond. We explain in this paper that, in our opinion, this assumption (which is also present in Rouse) is the origin of the failures of these models to describe the data correctly. The irony is that de Gennes [6] used the term “scaling concepts” in the title of his book on polymer physics [6], which resonates, but in a different context, with our definition of a scale of the basic unit that participates in the deformation process in our dissipative statistical approach. The difference is that our model not only defines the scale, in fact several “dynamic scales”, but also determines the coupling and the modulation between these cooperative scales [20]. For instance, In our Cross-Dual-Phase explanation of entanglements, we make reference to a “network of strands” to describe the cooperative interactive process resulting in the “entanglement phase”. We refer to a basic unit of deformation, the Dual-conformer, that participates in the evolving cooperative motion of a phase-wave responding to deformation as an open dissipative system [20]. We must define mathematically what “evolving cooperation” means, how many dual-conformers dynamically cooperate in an active strand at any instant, how many strands are active and how many relax, and where the cooperative dual-conformers are located: on a single chain or on several chains. The physics of dealing with all the chains at once in the statistics, redefining the coupling between the covalent and the inter-molecular interactions, is the model that we have adopted to describe the deformation of polymer melts and solids, above T_g_ and below T_g_ [2,19,20]. The theory not only addresses the interaction between the conformers of a single chain to assume the shape of a macro-coil (which can be deformed), but also defines why entangled macro-coils exhibit the response of a network of active strands when all the chains participate cooperatively in the deformation process. The dissipative dynamic coupling between the deformation of a conformer, of a macro-coil, and of a network of strands is quantitatively described. The new model explains the influence of chain molecular weight to predict a change in behavior below and above a critical molecular weight (M_e_), in other words it proposes a new understanding of “entanglements” and their influence on the dynamic melt properties G′(ω,T) and G″(ω,T) and the normal stresses. It predicts shear-thinning and strain softening in shear mode, and strain-hardening in extensional mode. It also successfully describes the transitional behavior at T_g_, from a solid-like to a liquid-like behavior, also predicting the existence and the characteristics of the Boyer’s T_LL_ upper melt transition temperature (the end of dissipative modulation). Finally, the theory addresses the stability (or the strain-induced lack of stability) of the Cross-Dual Phase entanglement network [20].

The theoretical assumptions of the new model and the quantitative descriptions it generates constitute a whole new understanding of the viscoelastic properties of polymers that could be considered the premises of a new paradigm in that field of physics. We would like to close by quoting Buckminster Fuller who once said:


*“In order to change an existing paradigm you do not struggle to try and change the problematic model. You create a new model and make the old one obsolete.”*
—Richard Buckminster Fuller

## Figures and Tables

**Figure 1 polymers-15-04309-f001:**
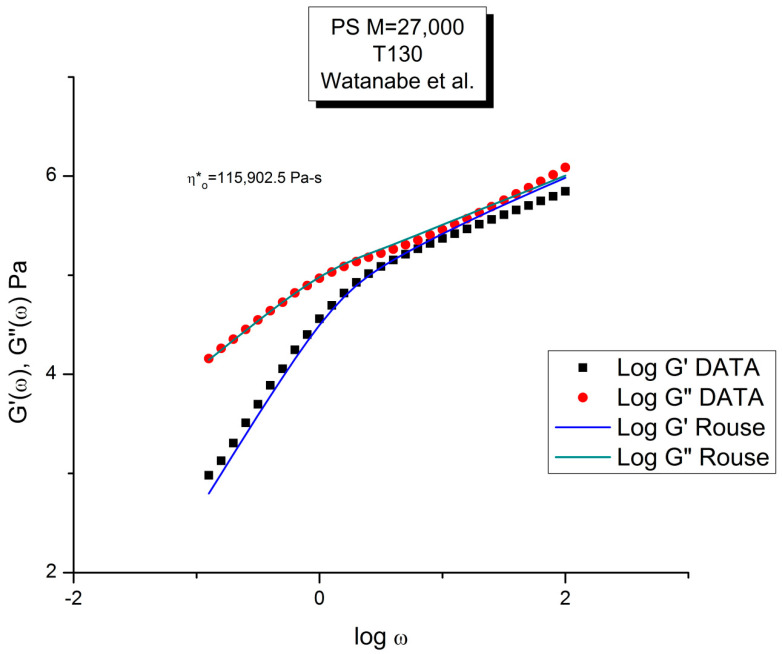
Comparison of log G′(ω), G″(ω) vs. logω for Data of Matsumiya and Watanabe [22] and the predictions of the Rouse model pursuant to Equations (1)–(4).

**Figure 2 polymers-15-04309-f002:**
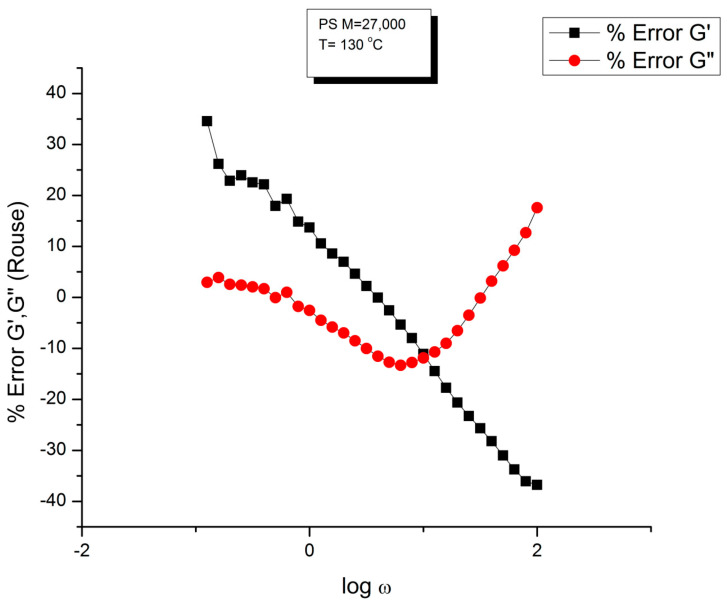
% error G′(ω) and G″(ω) with respect to Rouse simulation in Figure 1 plotted against ω.

**Figure 3 polymers-15-04309-f003:**
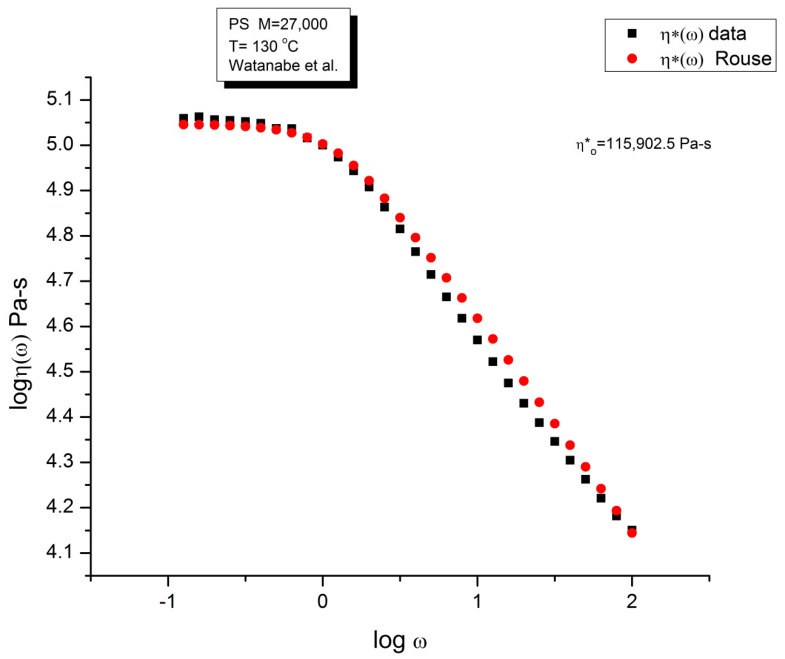
Comparison of the Rouse model prediction of log η*(ω) vs. logω and the data of Matsumiya and Watanabe [22] for PS (M = 27,000) at T = 130 °C (same data as in Figure 1).

**Figure 4 polymers-15-04309-f004:**
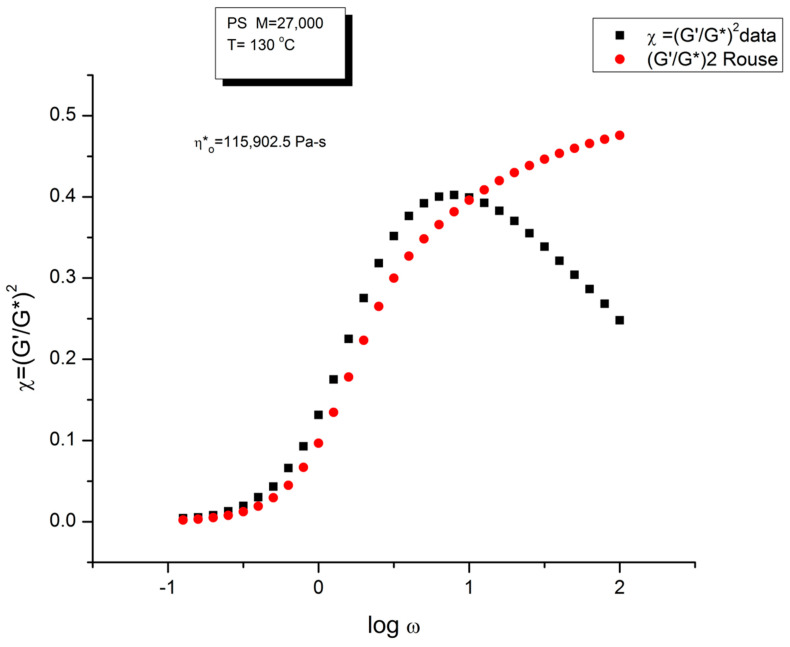
Comparison of χ = (G′/G*)^2^ vs. logω for the data in Figure 1, Figure 2 and Figure 3 and the predictions of the Rouse model pursuant to Equations (1)–(4).

**Figure 5 polymers-15-04309-f005:**
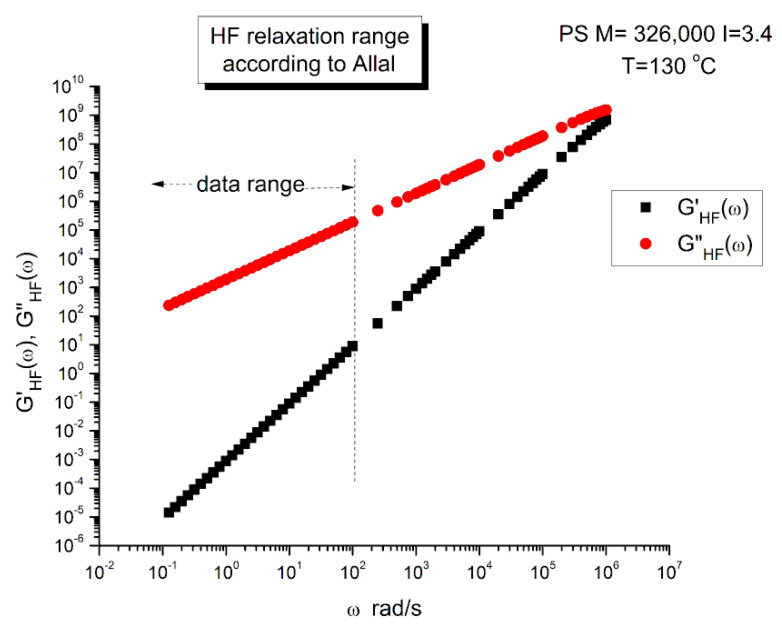
Calculated high frequency (HF) moduli, G′_HF_(ω), G″_HF_ (ω) vs. logω, pursuant to Allal [30], Equations (5) and (6), using the molecular parameters provided by Leonardi in [31] for a PS specified in the graph.

**Figure 6 polymers-15-04309-f006:**
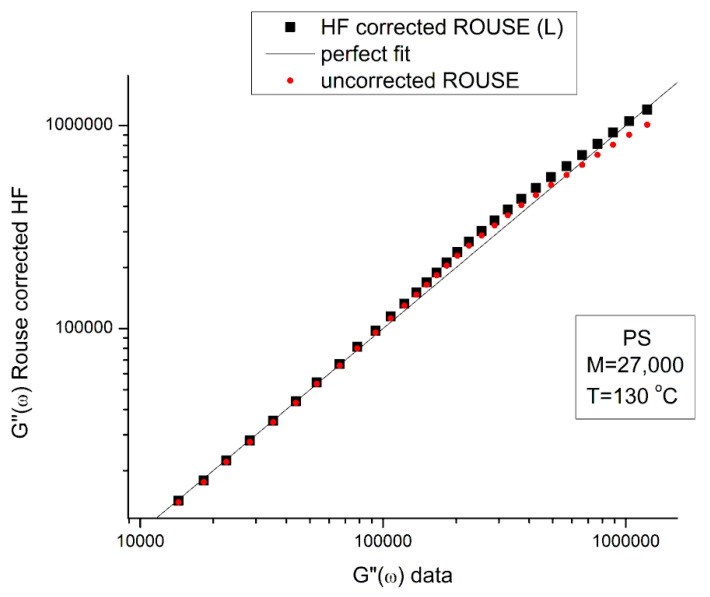
Plot of simulated G″(ω) using Rouse and the HF corrections from Figure 5 against G″(ω) for the data of Matsumiya and Watanabe, in Figure 1. The red dots apply to the uncorrected Rouse Equations (1)–(4) and the black squares to the corrected G″(ω) after adding G″_HF_(ω) calculated from Equations (5) and (6). The straight line is Y = X, assuming a perfect fit.

**Figure 7 polymers-15-04309-f007:**
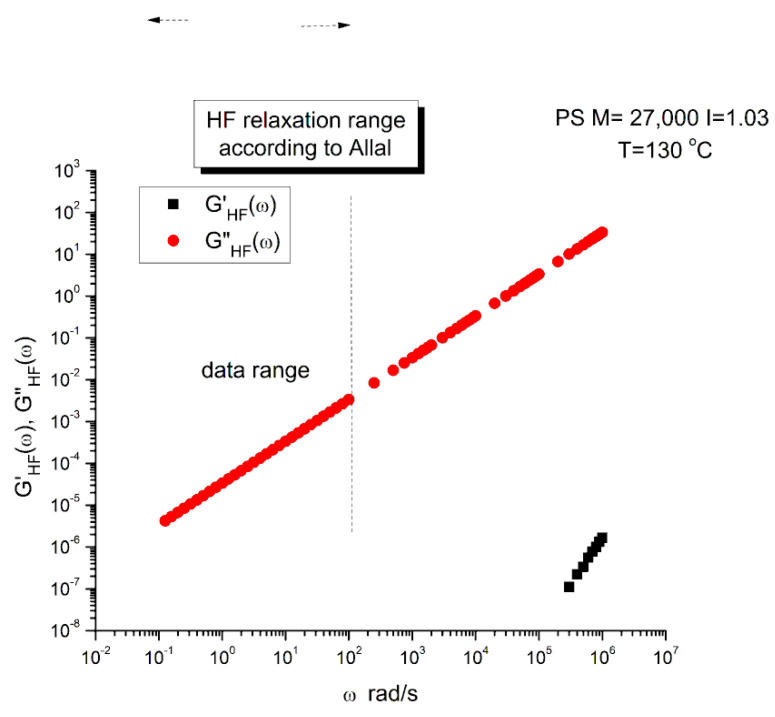
Calculated high frequency (HF) moduli, G′_HF_(ω), G″_HF_ (ω) vs. log ω, pursuant to Allal [30], Equations (5) and (6), using the molecular parameters provided by Majeste in [23] for a PS with the specifications of the Matsumiya and Watanabe’s sample.

**Figure 8 polymers-15-04309-f008:**
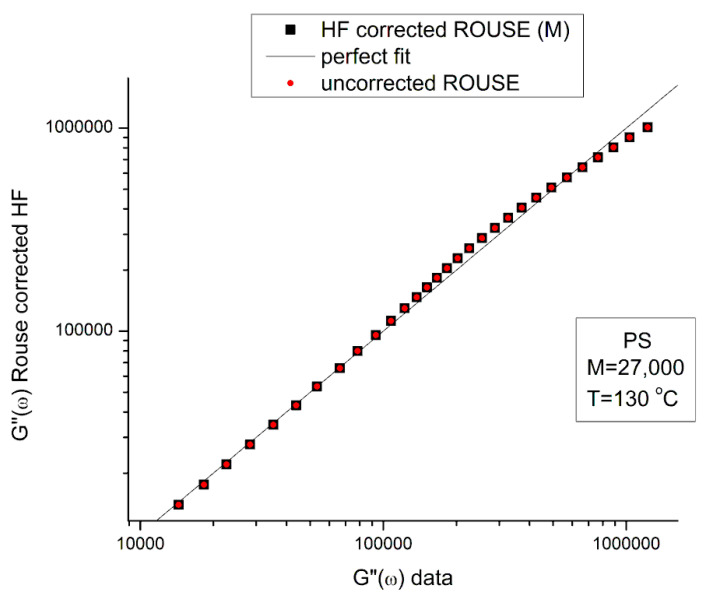
Plot of simulated G″(ω) using Rouse and the HF corrections from Figure 7 against G″(ω) for the data of Matsumiya and Watanabe. in Figure 1 The red dots apply to the uncorrected Rouse Equations (1)–(4) and the black squares to the corrected G″(ω) after adding G″_HF_(ω) calculated from Equations (5) and (6). The straight line is Y = X, assuming a perfect fit.

**Figure 9 polymers-15-04309-f009:**
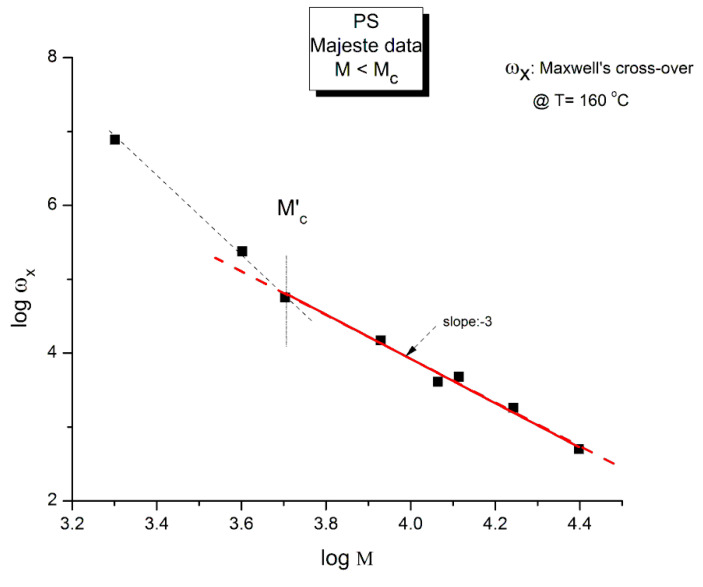
Logω_x_ vs. log M for un-entangled PS samples [23].

**Figure 10 polymers-15-04309-f010:**
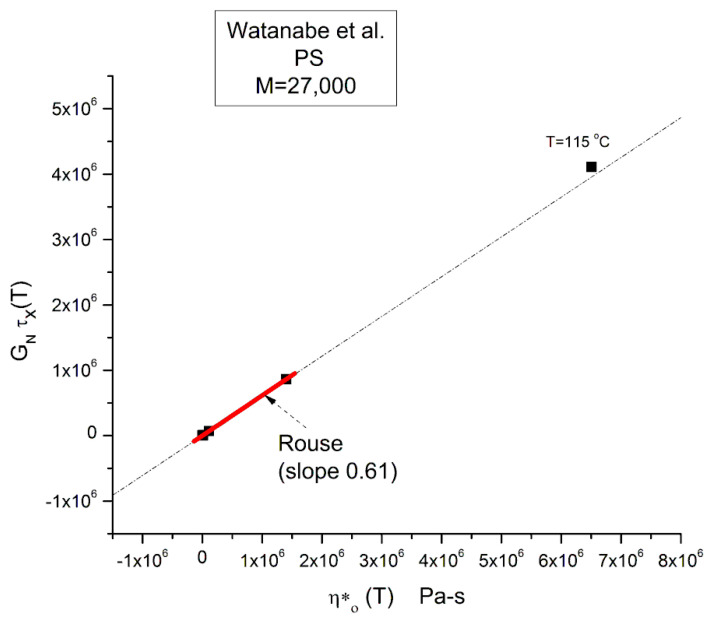
Validation of Rouse Equation (3) between G_N_, τ_x_ and n*_o_ in the Newtonian region using the data of Matsumiya and Watanabe [22] for PS with M = 27,000.

**Figure 11 polymers-15-04309-f011:**
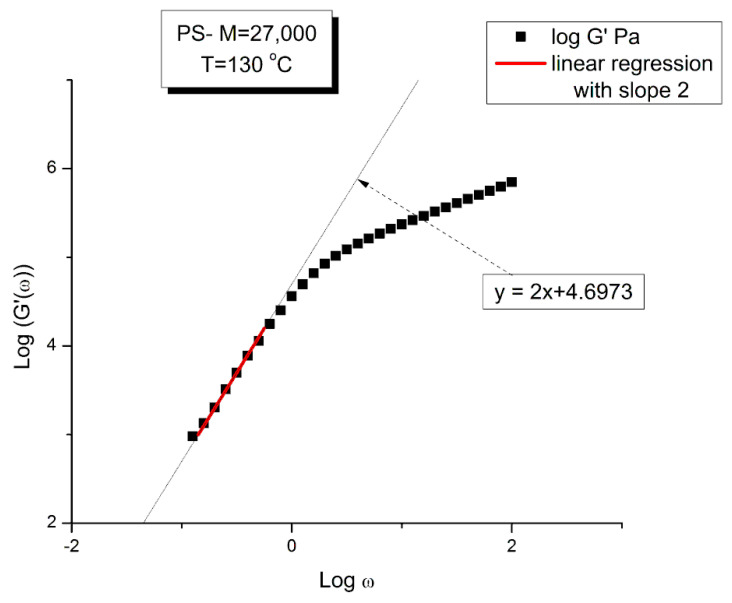
Testing the Rouse equation in the Newtonian regime from a G′(ω) point of view pursuant to Equation (7a): A plot of log G′(ω) vs. logω is fitted in the low ω range with a Maxwell’s straight line of slope 2. The value of G_N_ is derived from the fit.

**Figure 12 polymers-15-04309-f012:**
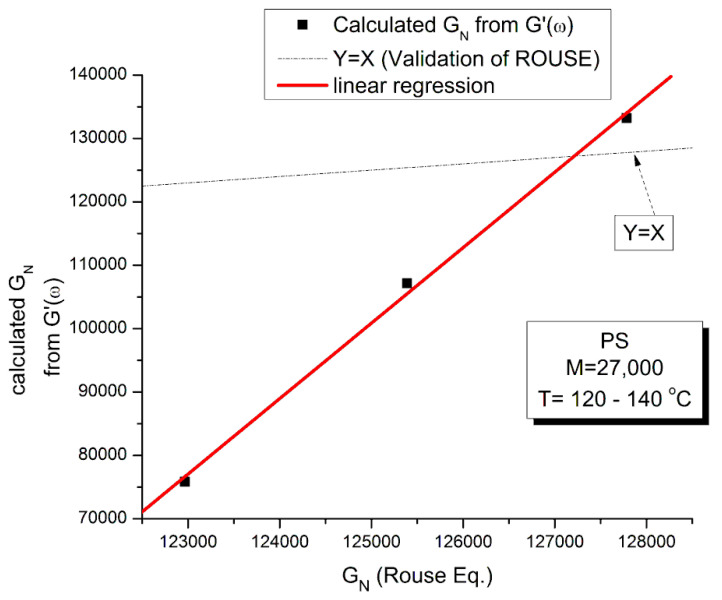
Invalidation of Rouse Equations (1)–(4) in the Newtonian regime from a G′(ω) point of view pursuant to Equation (7a): the value of G_N_ is totally different from the Rouse value determined from G″(ω).

**Figure 13 polymers-15-04309-f013:**
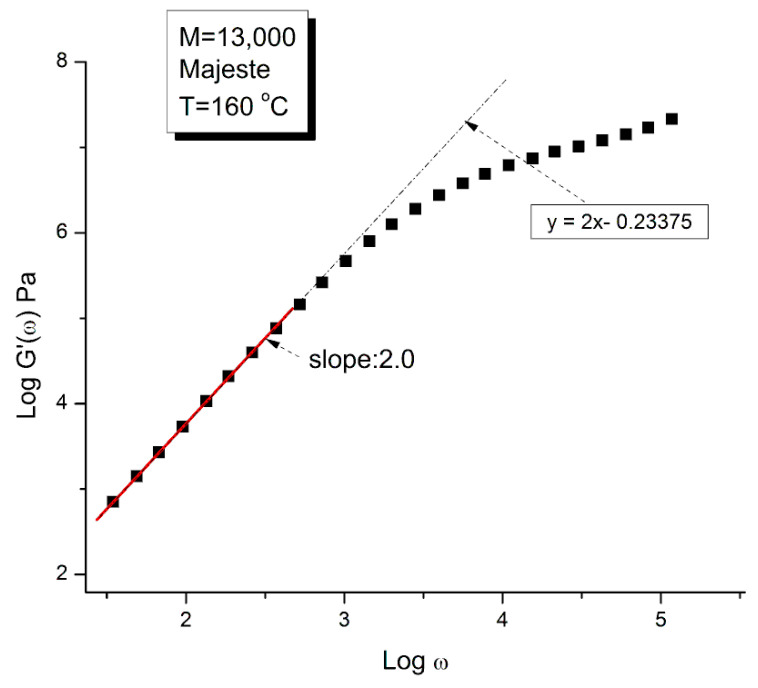
Same testing of the Rouse equation as in Figure 11, here applied to Majeste’s data [23] at T = 160 °C, M = 13,000.

**Figure 14 polymers-15-04309-f014:**
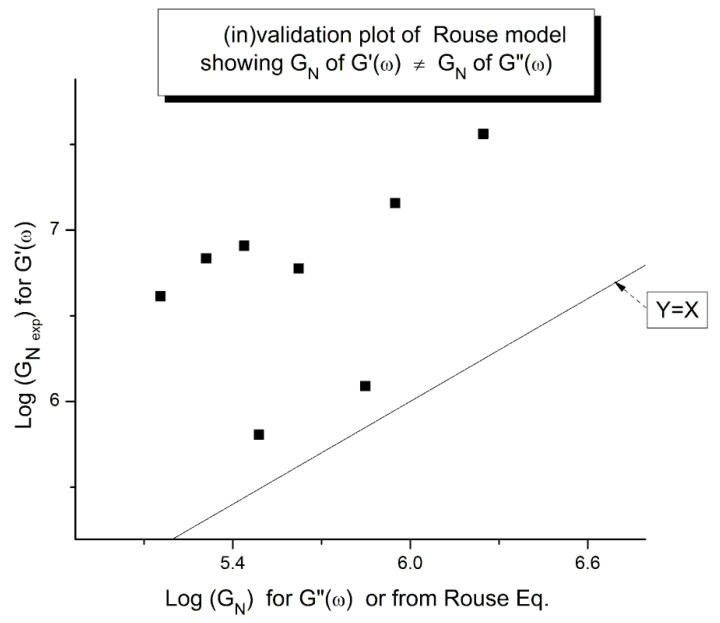
Comparison of the G_N_ values calculated from G′(ω) and G″(ω) in the Newtonian region using the Rouse Equations (1)–(6). The points should be on the Y = X line for validation of the Rouse model.

**Figure 15 polymers-15-04309-f015:**
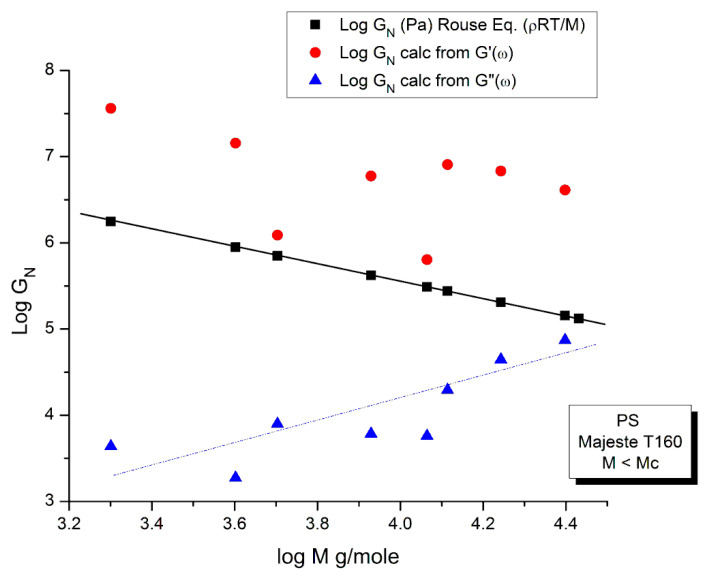
Comparing the value of G_N_ calculated from G′(ω), G″(ω) and from the Rouse’s formula based on the rubber elastic theory.

**Figure 16 polymers-15-04309-f016:**
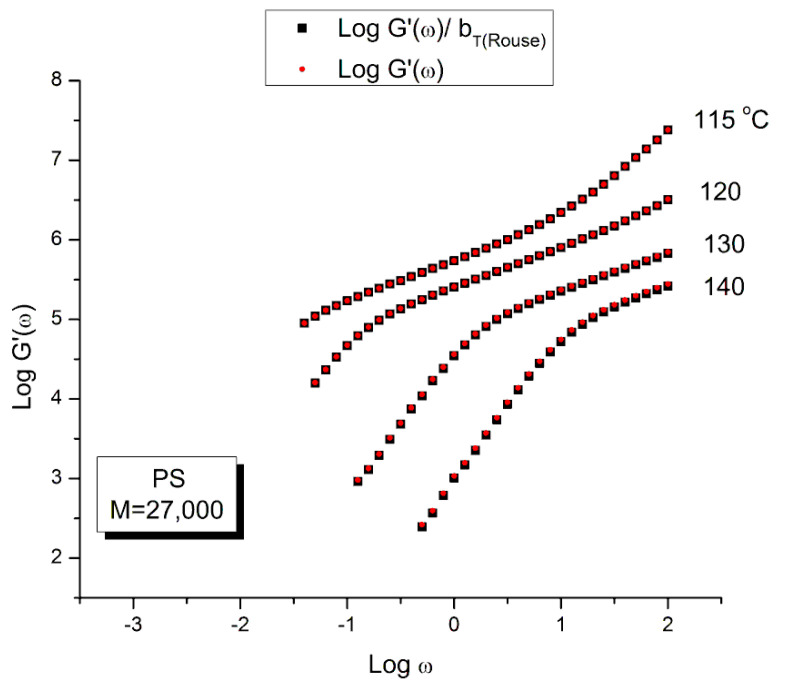
Comparison of log (G′/b_TRouse_) and log (G′) vs. log ω at various temperatures for Matsumiya and Watanabe PS = 27,000 [22].

**Figure 17 polymers-15-04309-f017:**
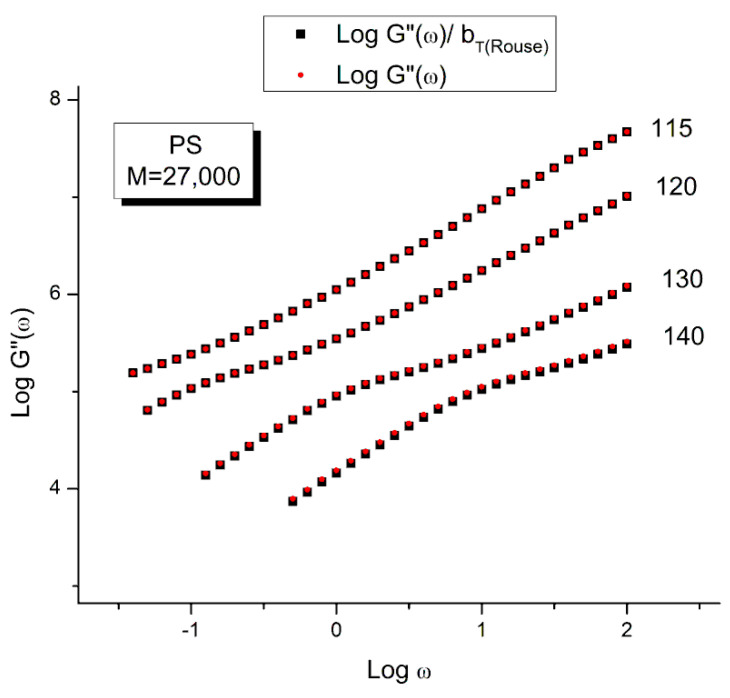
Comparison of log (G″/b_TRouse_) and log (G″) vs. log ω at various temperatures for Matsumiya and Watanabe PS = 27,000 [22].

**Figure 18 polymers-15-04309-f018:**
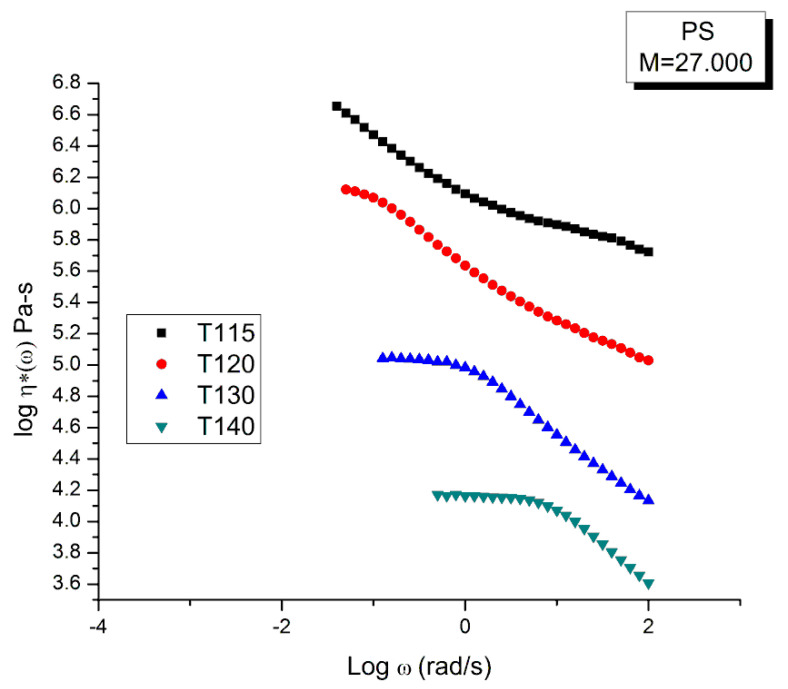
log η*(ω) vs. log ω at 4 temperatures for PS M = 27,000. Raw data [22].

**Figure 19 polymers-15-04309-f019:**
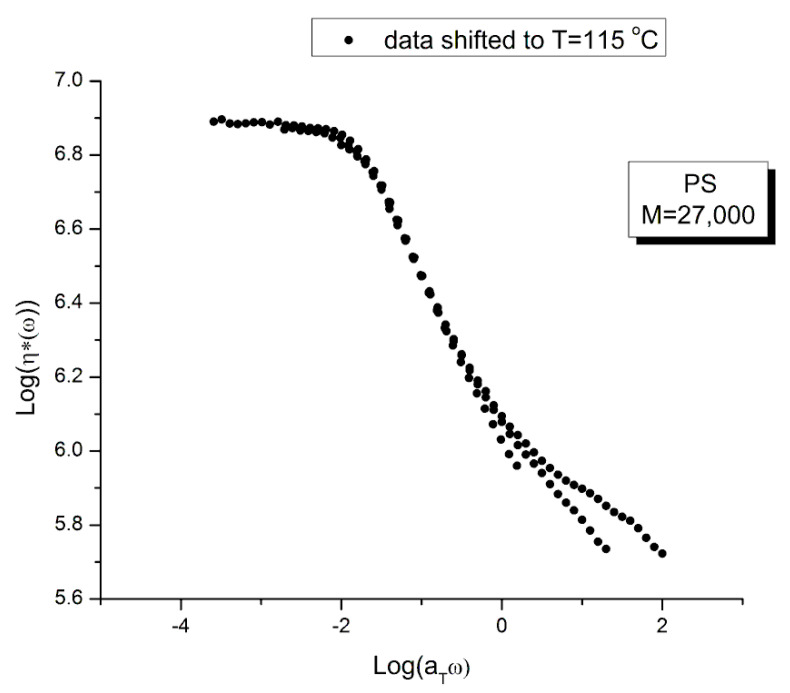
Viscosity mastercurve of log η*(ω) vs. log $a_{T}\omega$ after horizontal shifting of the curves of Figure 18 onto the T_1_ = 115 °C curve.

**Figure 20 polymers-15-04309-f020:**
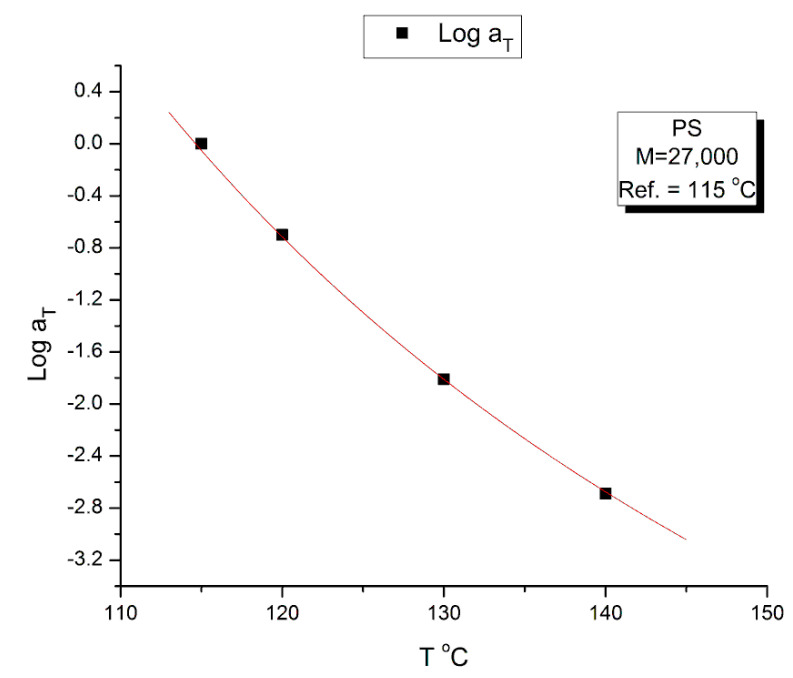
log a_T_ vs. T for shifting the curves of Figure 18 onto the reference temperature T_1_ = 115 °C.

**Figure 21 polymers-15-04309-f021:**
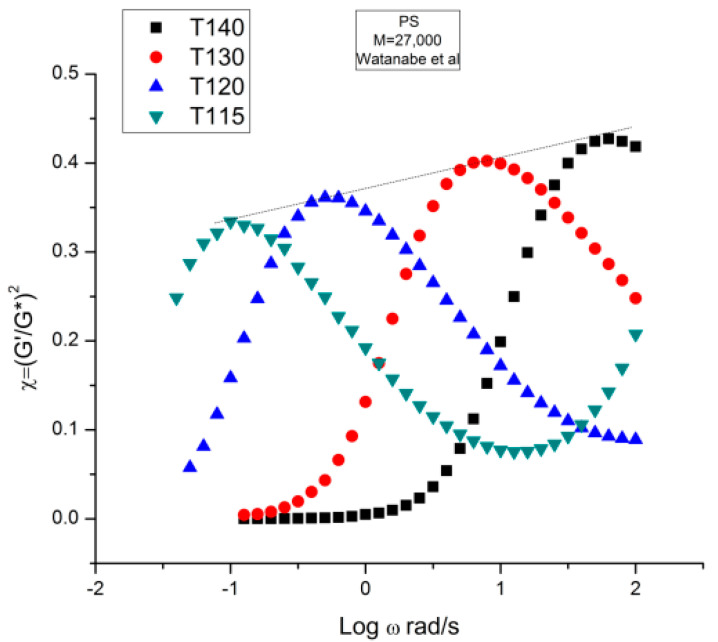
χ = (G′/G*)^2^ vs. log ω for the Matsumiya and Watanabe [22] PS = 27,000 at 4 temperatures.

**Figure 22 polymers-15-04309-f022:**
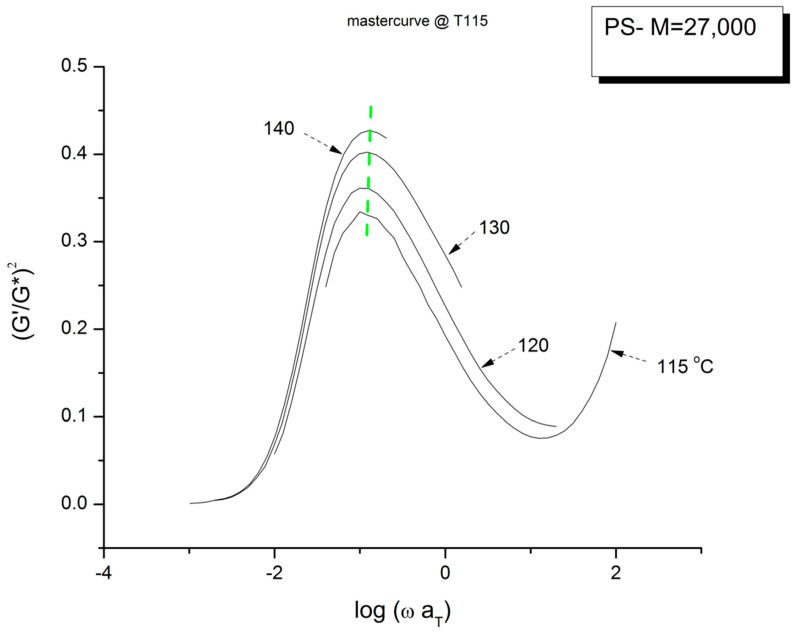
Mastercurve at T = 115 °C obtained after shifting horizontally the data of Figure 21 by the log a_T_ of Figure 20, i.e., the values that were used to shift the viscosity curves of Figure 18 to obtain the mastercurve of Figure 19.

**Figure 23 polymers-15-04309-f023:**
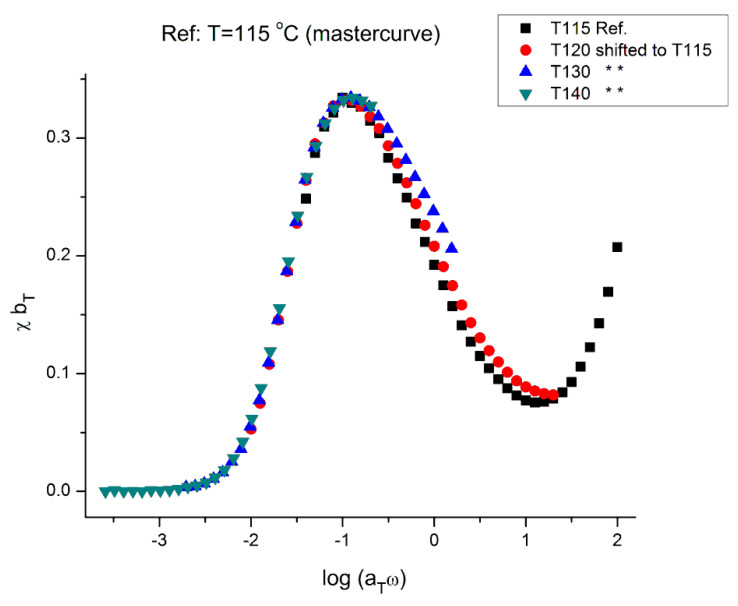
(b_T_χ) plotted against log (a_T_ ω) at T_1_ = T_Ref_ = 115 °C.

**Figure 24 polymers-15-04309-f024:**
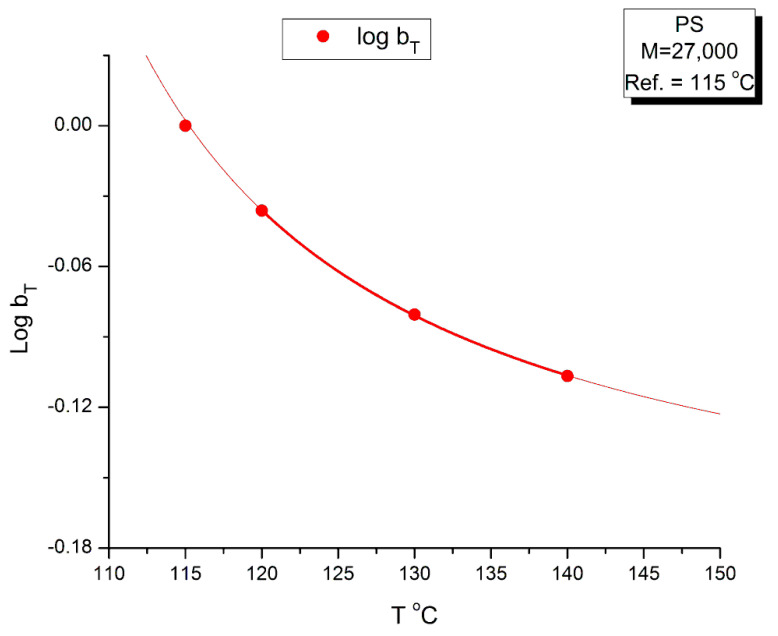
log b_T_ of the vertical shift factor in Figure 23 plotted against T showing an hyperbolic fit.

**Figure 25 polymers-15-04309-f025:**
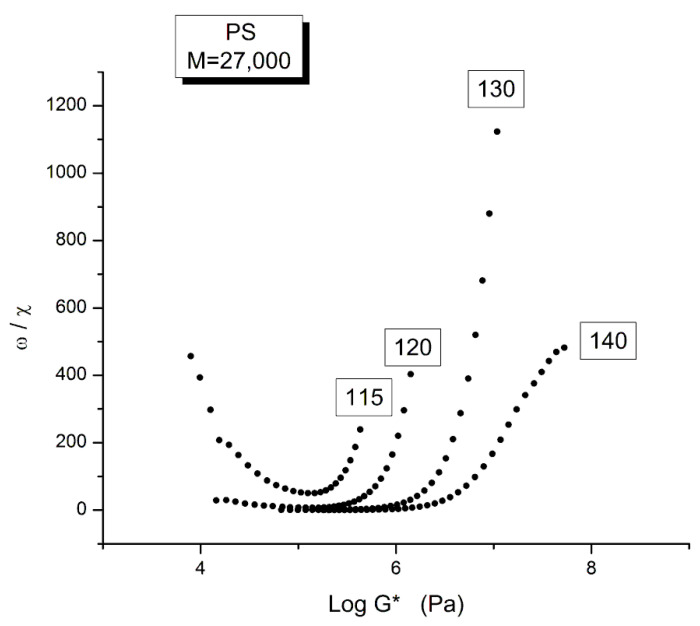
Plot of ω′ = ω/χ against log G* at the 4 temperatures of the Matsumiya and Watanabe. data [22] demonstrates the presence of the T_g_ + 23 °C transition at T~115 °C. See text.

**Figure 26 polymers-15-04309-f026:**
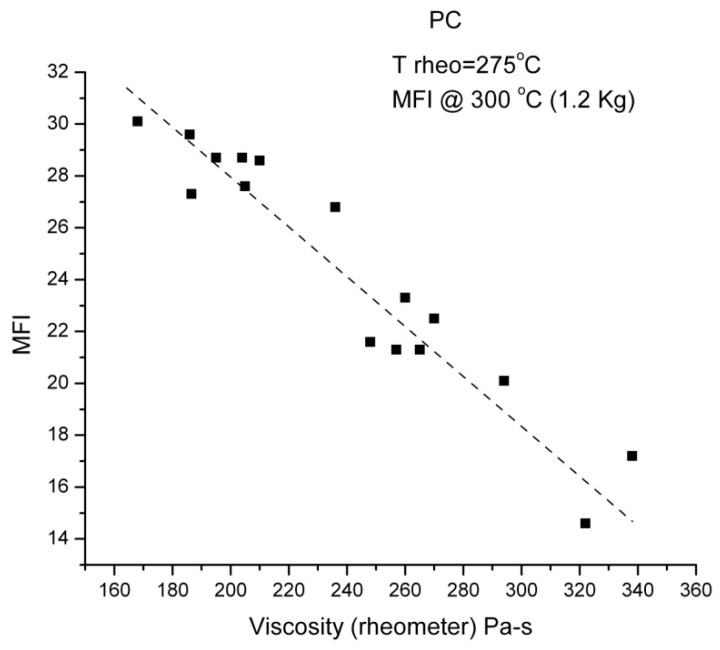
Signature of “Sustained-Orientation”: MFI value found for pellets made out of a melt prepared by Rheo-Fluidification treatment are linearly correlated to the value of viscosity measured by the in-line viscometer at the exit of the Rheo-Fluidizer.

**Figure 27 polymers-15-04309-f027:**
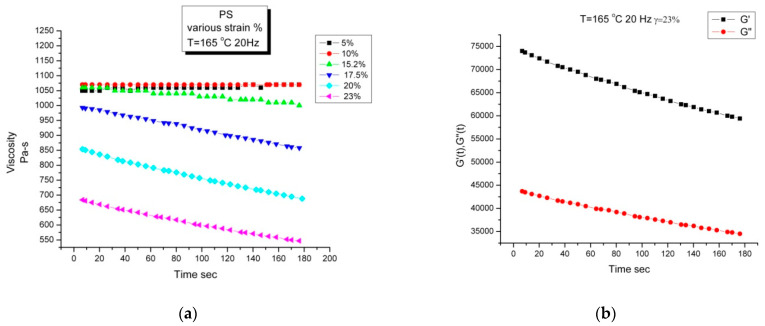
(**a**) Viscosity (Pa-s) vs. Time (s) during successive time sweep sequences of 3 min each at 165 °C, 20 Hz for a PS sample in a dynamic rheometer. The strain is increased at the beginning of each sequence as shown in the inset. Viscosity is calculated from G′(t) and G″(t). (**b**) Details of Figure 27a regarding the strain = 23% sequence. This graph shows the decay of G′ and G″ with time.

**Figure 28 polymers-15-04309-f028:**
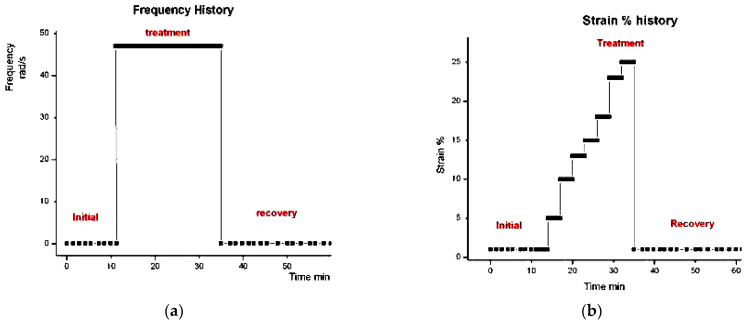
(**a**) Frequency history steps (plotted against time) for the sample in the dynamic rheometer. (**b**) Strain % history steps (plotted against time) for the sample in the dynamic rheometer of Figure 28a.

**Figure 29 polymers-15-04309-f029:**
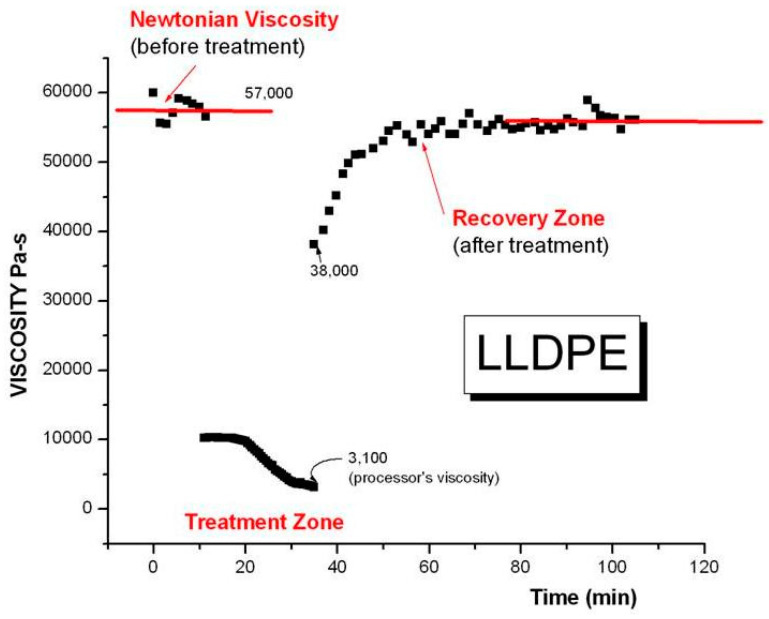
Dynamic viscosity vs. time for the 3 steps of Figure 28a,b.

**Table 1 polymers-15-04309-t001:** Working variables to test the Rouse model (see text): “Rau” = ρ; “tau” = τ.

	T ^o^C	Log τ_x_ τ_x_ =1/ω_x_	Log η*_o_ Pa-s (Eq. (4))	ρ (g/cc) melt density	G_N_ (Pa) PS 27,000 ρRT/M	G_N_ x τ_x_	η*_O_ Pa-s	I′_M_ See text	G_N_ calc. from G′(ω)
1	115	1.52824	6.8135	1.01854	121,744.4	4.11 × 10^6^	6.51 × 10^6^		-
2	120	0.84678	6.14749	1.01568	122,966.2	864,105.4	1.40 × 10^6^	6.78861	75,843.4
3	130	−0.27389	5.04994	1.01	125,389.1	66,736.86	112,187.5	4.6973	107,141.6
4	140	−1.15712	4.18292	1.00439	127,785.1	8899.472	15,237.84	3.02546	133,217.8

**Table 2 polymers-15-04309-t002:** Working variables to test the Rouse model (see text): “Rau” = ρ; “tau” = τ.

	M g/mole	ρ g/cm^3^	Log G_N_ calc from ρRT/M	I’_M_ (see text)	ω_x_(M) Maxwell rad/s	Log G_N_ calc from G′(ω)	Log η*_o_ (Pa-s) [23]	Log G_N_ calc from G″(ω)
1	2000	0.98129	6.24709	−6	7.73 × 10^6^	7.56106	−3.03205	3.6399
2	4000	0.98803	5.94903	−3.38286	238404.6	7.15664	−1.88678	3.27439
3	5050	0.98945	5.84842	−3.19938	56491.44	6.08946	−0.6361	3.89974
4	8500	0.99164	5.62326	−1.35373	14874.16	6.77601	−0.17089	3.78539
5	11600	0.99251	5.4886	−1.20429	4096.45	5.8054	0.36388	3.76014
6	13000	0.99276	5.43922	−0.23375	4769.655	6.90809	0.83051	4.29285
7	17500	0.9933	5.31037	0.53143	1815.841	6.83446	1.6037	4.64663
8	25000	0.99378	5.15567	1.42833	501.593	6.6139	2.38771	4.87192

**Table 3 polymers-15-04309-t003:** Horizontal and vertical shift factors, a_T_ and b_T_, respectively, for the superposition of the rheological data of Matsumiya and Watanabe [22].

Temperature °C	Log a_T_ T _Ref_ = 115	Log b_T_ (from χ) T _Ref_ = 115
115	0	0
120	−0.7	−0.03617
130	−1.81	−0.08049
140	−2.69	−0.10672

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
