# Peer review of "The Challenges Facing the Current Paradigm Describing Viscoelastic Interactions in Polymer Melts"

_polymers, 2023, doi:10.3390/polym15214309_

Round 1
Reviewer 1 Report
The paper from Ibar critically reviews the viscoelastic interactions occurring in polymer melts, on the basis of the current different rheological models that describe (or even fail to describe) the rheological behavior of macromolecular chains in the molten state (i.e., the deformation of a polymer melt in shear mode), according to their structure, MW, etc... The paper is quite well written and presents a critical assessment of the models already applied for the interpretation of the macromolecular dynamics in the molten state.
Some comments and suggestions are listed as follows:
- The manuscript provides the point of view of the Author, who critically examines the currently existing models for interpreting the molecular dynamics of polymer melts, highlighting their limitations and weaknesses. Compared to other published material, the manuscript could be considered quite original
- The methodology proposed by the Author seems consistent and does not require specific improvements. No further controls should be considered
- the Abstract is really too long and should be rewritten, significantly shortening it
- Line 161: please replace "Compare" with "Comparison" (this is also suggested for all the other figures throughout the text, where "Compare" is present)
- Tables 1-3: please provide a short description in the caption
- lines 1084-1092 ("Following such radical conclusive statements regarding the merit of the reptation model, it is important to verify the credibility of the authors of the study: they are all scientists working at the most reliable and prestigious international institutions in the world: the 1,2Oak Ridge National Laboratory, Oak Ridge, Tennessee (USA), the 4Center 1088 for Neutron Research, National Institute of Standards and Technology, in Gaithersburg, Maryland, the Department of Polymer Science of the University of Akron in Ohio(USA), the Institut Laue‐Langevin, in Grenoble France, and the Department of Chemical and Biomolecular Engineering, University of Delaware (USA). In my view, the competence of the authors is impeccable.") should be deleted: this is the author's opinion and is far from being a scientific statement. In other words, working at MIT (USA) does not imply to be the best researcher just because of the high reputation of MIT...
- The conclusions are consistent with the evidence and arguments presented in the manuscript and address the main question posed
- References: it seems there's an excess of self-citations.
Apart from the other comments I made, I would stress that the manuscript has a big issue with self-citations, and therefore the Author should balance the References. Besides, even though I did not mention this in my review report, I’m not fully convinced that the manuscript classification chosen by the Author (i.e., article) is correct, as there’s a huge part of the manuscript that reviews and comments the already existing literature on the topic. In this view, maybe the manuscript should be considered as a review paper.
Author Response
Dear Reviewer,
Thank you for reviewing my submission and your comments. I have taken all your remarks into consideration and modified the submission accordingly
I have made the Abstract shorter.
I accept your suggestion to consider the paper a REVIEW and have modified the way it is designated : a review instead of an article.
For the self-references: I agree there were a lot and have deleted Refs 71 to 85, modifying the text correspondingly. The problem with the introduction of new ideas is that the papers are often ignored and not cited by others, and the way to make the work known and to not repeat its content in new publications is to refer to it so the reader can check the content as further support for the new ideas. I have reconsidered each of my references and decided to delete it or keep it as self-references after consideration of the following: is the content of the self-cited reference adding something to the issue debated in the present paper. If I could quote it differently by only referring to my books, then I deleted it. I am afraid that those remaining (still a lot, I regret to admit) are probably necessary for a serious reader who needs to be convinced by evidence not rhetoric. All the other issues have been taken of.
Thank you for your input, I appreciate it.
Jean Pierre Ibar

Reviewer 2 Report
Comment for polymers-2616065 is listed in the attachment.

None.
Author Response
Thank you for your review. I have addressed all the issues you have raised and modified the text accordingly.
The only question that remained might have been related to the bad transcription of the equations in your 1st version since I could not address your concerns about Eq (1) or (6).
I have highlighted all the modifications in yellow.
I am sending the pdf version so you have the correct conversion of the equations.
I thank you for your time making valuable revisions possible..
I appreciate your input.
Best regards,
Jean Pierre Ibar
Best regards,

Reviewer 3 Report
There are fewer and fewer experts in rheology nowadays. I saw many researchers in the polymer process community rely on rheological data in their study, but blindly applying theories without really knowing why or just simply following other works. I believe all of them can find something valuable from this review paper. I suggest publication in the present form.
Author Response
THANK YOU!
I appreciate your time reading my manuscript and thank you for your nice comment.
Best regards.
Jean Pierre Ibar
Round 2
Reviewer 1 Report
The author revised the manuscript according to the reviewer's comments. Therefore, the manuscript is now suitable for publication.